# Enhancing Personal Decentralized Federated Learning through Model Decoupling

## Abstract

Personalized Federated Learning (FL) aims to produce many local personalized models rather than one global model to encounter an insurmountable problem – data heterogeneity in real federated systems. However, almost all existing works have to face central communication burdens and the risk of disruption if the central server fails. Only limited efforts have been made without a central server but they still suffer from high local computation, catastrophic forgetting, and worse convergence due to the full model aggregation process. Therefore, in this paper, we propose a PFL framework through model decoupling called DFedMDC, which pursues robust communication and better model performance with a convergence guarantee. It personalizes the "right" components in the modern deep models by alternately updating the shared and personal parameters to train partially personalized models in a peer-to-peer manner. To further promote the shared parameters aggregation process, we propose DFedSMDC via integrating the local Sharpness Aware Minimization (SAM) optimizer to update the shared parameters. Specifically, it adds proper perturbation in the gradient direction to alleviate the shared model inconsistency across clients. Theoretically, we provide convergence analysis of both algorithms in the general non-convex setting with partial personalization and SAM optimizer for the shared model. We analyze the ill impact of the statistical heterogeneity $\delta^2$, the smoothness $L_u, L_v, L_{uv}, L_{vu}$ of loss functions, and communication topology $(1 - \lambda)$ on the convergence. Our experiments on several real-world data with various data partition settings demonstrate that (i) partial personalized training is more suitable for personalized decentralized FL, which results in state-of-the-art (SOTA) accuracy compared with the SOTA PFL baselines; (ii) the shared parameters with proper perturbation make partial personalized FL more suitable for decentralized training, where DFedSMDC achieves most competitive performance.

## 1 Introduction

Federated Learning (FL) is an emerging technique for preserving privacy and reducing the communication cost in training machine learning models without requiring raw data sharing. Generally, one central server is needed to aggregate the model from each client and gather an average model for the whole system. However, data heterogeneity among participating clients makes it hard to achieve a satisfactory performance for all clients using one average model. Personalized Federated Learning (PFL) is thus proposed to achieve several local personalized models for each client via focusing the shift from the global average model on the server to the local personalized models on the clients.

In the context of PFL, existing works with a central server can be roughly divided into five categories: parameter decoupling (Arivazhagan et al., 2019; Collins et al., 2021; Oh et al., 2021), knowledge distillation (Li and Wang, 2019; Lin et al., 2020; He et al., 2020), multi-task learning (Huang et al., 2021; Shoham et al., 2019), model interpolation (Deng et al., 2020; Diao et al., 2020) and clustering (Ghosh et al., 2020; Sattler et al., 2020). All of these methods learn a global model explicitly or implicitly through the central server, then achieve personalization by analyzing the relationship between global and local models. However, all communication processes in these methods need the central server to aggregate local models, which may cause a quite large communication burden on the server side. Moreover, the system may suffer the risk of disruption if the central server fails. There only exists limited efforts focusing on personalized model aggregation in a peer-to-peer

manner without a central server (Jeong and Kountouris, 2023; Sadiev et al., 2022; Dai et al., 2022). Though it avoids central failure, the choice of the correct clients or parameters involves a large amount of computation and communication costs. Besides, the full model aggregation may lose unique information for each client due to mixing the linear classifier, which will lead to catastrophic forgetting and worse convergence. All in all, we are trying to explore:

> *Can we design a personalized algorithm, where there is simultaneously robust communication and better model performance with convergence guarantee?*

To answer this question, we propose DFedMDC, which decouples models as a mixture of a shared feature representation part and a personalized linear classifier part and optimizes them alternatively in a peer-to-peer manner. Instead of calculating to find the right neighbors or parameters for aggregation, DFedMDC only directly shares and aggregates the "right" part with their neighbors and allows the private linear classifier to better adapt to their local data. Specifically, clients aggregate the received shared parameters and optimize the personalized part and the shared part alternatively. Then it exchanges the updated shared parameters with their neighbours directly without a central server. To the best of our knowledge, we are the first to explore partial personalization in a peer-to-peer manner and overcome the lose unique information of each client. Where we decompose each local model and only average a shared part with its neighbors of each client. Furthermore, we propose an enhanced version of DFedMDC, called DFedSMDC, which integrates a local SAM optimizer to update the shared parameters. Specifically, it searches for the shared parameters with uniformly low loss values by adding proper perturbation in the direction of the gradient, thereby promoting the process of local model aggregations in each client (see Section 3).

Theoretically, we present the non-trivially convergency analysis for both DFedMDC and DFedSMDC algorithms in the general non-convex setting (see Section 4), which can analyze the ill impact of the statistical heterogeneity $\delta^2$, the smoothness $L_u, L_v, L_{uv}, L_{vu}$ of loss functions, and communication topology $(1 - \lambda)$ on the convergence with partial personalization and SAM optimizer for the shared model. Empirically, we conduct extensive experiments on CIFAR-10, CIFAR-100, and Tiny-ImageNet datasets in non-IID settings with different data partitions, such as Dirichlet settings with various $\alpha$ and pathological settings with various limited classes in each client. Experimental results confirm that our algorithms can achieve competitive performance relative to many SOTA PFL baselines (see Section 5). In summary, we provide a comprehensive study focusing specifically on partial model personalization in a peer-to-peer manner. Our main contributions lie in four-fold:

- Considering the over-fitting in local training but catastrophic forgetting in global aggregation due to a fully personalized model, we seek out a suitable personalization federated learning method with robust communication and fast convergence and propose DFedMDC via alternately updating the shared part and personal part in a peer-to-peer manner.
- To further improve the model aggregation, we propose DFedSMDC, which integrates the local SAM optimizer into the shared parameters to enhance the flattenness and robustness of the shared parts.
- We provide convergence guarantees for the DFedMDC and DFedSMDC methods in the general non-convex setting with peer-to-peer partial participation in PFL.
- We conduct extensive experiments on realistic data tasks with various data partition ways, evaluating the efficacy of our algorithms compared with some SOTA PFL baselines.

## 2 RELATED WORK

**Personalized Federated Learning (PFL).** Compared to the FL pursuing a more robust global model for clients' non-iid distributions, the PFL aims to produce the greatest personalized models for each client. From the perspective of learning personalized models, there mainly exist five categories of methods: model decoupling (Arivazhagan et al., 2019; Collins et al., 2021; Oh et al., 2021), knowledge distillation (Li and Wang, 2019; Lin et al., 2020; He et al., 2020), multi-task learning (Huang et al., 2021; Shoham et al., 2019), model interpolation (Deng et al., 2020; Diao et al., 2020) and clustering (Ghosh et al., 2020; Sattler et al., 2020). More details can be referred to in (Tan et al., 2022). In this paper, we mainly focus on the model decoupling methods, which divide the model into a global shared part and a personalized part, also called *partial personalization*.

**Partial Personalization in FL.** Existing works demonstrate that partial model personalization can outperform of full model personalization with fewer shared parameters. Specifically, FedPer (Arivazhagan et al., 2019) uses the one global body with many local heads approach and only shares the body layers with the server. FedRep (Collins et al., 2021) learns the entire model sequentially with the head updating first and the body later, and only shares the body layers with the server. FedBABU (Oh et al., 2021) trains the global body with a fixed head for all clients and finally fine-tunes the personalized heads on the basis of the consensus body. Fed-RoD (Chen and Chao, 2021) leverages a global body and two heads, e.g., the generic head trained with class-balanced loss and the personalized head trained with empirical loss. FedSim and FedAlt in (Pillutla et al., 2022) provide the first convergence analyses of both algorithms in the general nonconvex setting with partial participation. Inspired by this, we provide the non-trivial convergence analysis on decentralized partial model personalization and deliver theoretical analysis at first combining partial personalization with various peer-to-peer communication networks and the SAM optimizer.

**Decentralized Federated Learning (DFL).** In DFL, the clients only connect with their neighbors and its goal is to make all local models tend to a unified model through peer-to-peer communication. Due to the participants having different hardware and network capabilities in the real federated system, DFL is an encouraging field in recent years (Beltrán et al., 2022; Kang et al., 2022; Li et al., 2022a; Nguyen et al., 2022; Wang et al., 2022; 2020; Yu et al., 2020). For some applications, BrainTorrent (Roy et al., 2019) is the first serverless, peer-to-peer FL approach applied to medical applications in a highly dynamic peer-to-peer FL environment, while DFedSAM (Shi et al., 2023b) integrates Sharpness Awareness Minimization (SAM) into DFL to improve the model consistency across clients. Similar to general FL methods such as (McMahan et al., 2017), we discuss the PFL methods in DFL considering both multi-step local iterations and various communication topologies.[1] Specifically, DFedAvgM (Sun et al., 2022) applies the multiple local iterations with SGD and quantization method to reduce the communication cost. Dis-PFL (Dai et al., 2022) customizes the personalized model and pruned mask for each client to further lower the communication and computation cost. KD-PDFL (Jeong and Kountouris, 2023) leverages knowledge distillation technique to empower each device so as to discern statistical distances between local models. The work in (Sadiev et al., 2022) presents lower bounds on the communication and local computation costs for this personalized FL formulation in a peer-to-peer manner. To reduce the central server's communication burden and the risk of disruption if the central server fails, in this work, we leverage a decentralized communication way to aggregate the shared model based on model decoupling.

## 3 METHODOLOGY

In this section, we define the problem setup for DFL and decentralized partial personalized models in PFL at first. After that, we present two algorithms: DFedMDC and DFedSMDC in PFL, which leverages the decentralized partial model personalization technique to generate better representation ability while achieving SOTA performance relative to many related PFL methods.

### 3.1 PROBLEM SETUP

**Decentralized Federated Learning (DFL).** We consider a typical setting of DFL with $m$ clients, where each client $i$ has the data distribution $\mathcal{D}_i$. Let $w \in \mathbb{R}^d$ represent the parameters of a machine learning model and $F_i(w; \xi)$ is the local objective function associated with the training data samples $\xi$. Then the loss function associated with client $i$ is $F_i(w) = \mathbb{E}_{\xi \sim \mathcal{D}_i} F_i(w; \xi)$. After that, a common objective of DFL is the following finite-sum stochastic non-convex minimization problem:

$$\min_{w \in \mathbb{R}^d} F(w) := \frac{1}{m} \sum_{i=1}^{m} F_i(w). \tag{1}$$

In the decentralized network topology, the communication between clients can be modeled as an undirected connected graph $\mathcal{G} = (\mathcal{N}, \mathcal{V}, \boldsymbol{W})$, where $\mathcal{N} = \{1, 2, \ldots, m\}$ represents the set of clients, $\mathcal{V} \subseteq \mathcal{N} \times \mathcal{N}$ represents the set of communication channels, each connecting two distinct clients, and

---

[1]In decentralized/distributed training, they also focus on peer-to-peer communication, but one-step local iteration is adopted, due to the gradient computation being more focused than the communication burden. More detailed related works in decentralized/distributed training are placed in **Appendix** A.

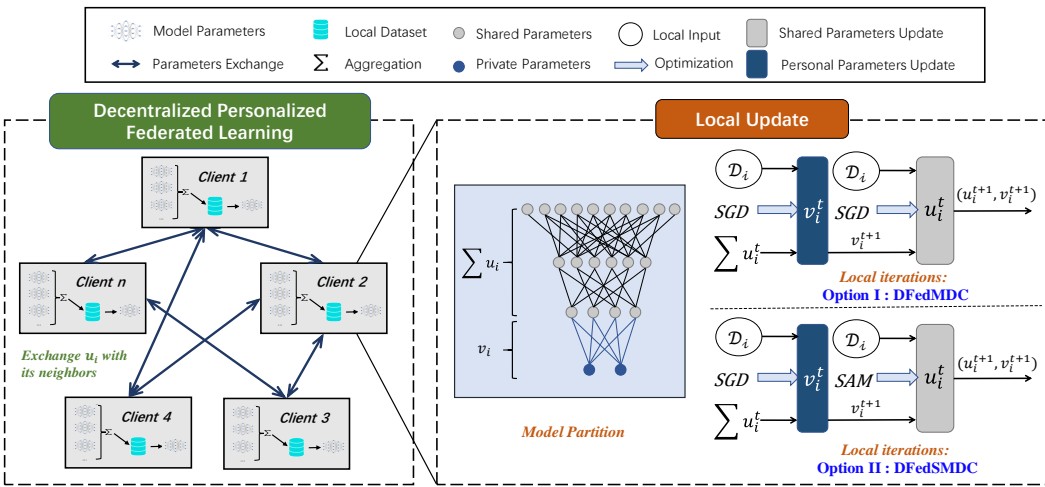

Figure 1: An overview of the proposed DFedMDC and DFedSMDC frameworks.

the gossip/mixing matrix $W$ records whether the communication connects or not between any two clients. As below, we present the definition of $W$:

**Definition 1** (The gossip/mixing matrix (Sun et al., 2022))*. The gossip matrix $\mathbf{W} = [w_{i,j}] \in [0,1]^{m \times m}$ is assumed to have these properties: (i) (Graph) If $i \neq j$ and $(i,j) \notin \mathcal{V}$, then $w_{i,j} = 0$, otherwise, $w_{i,j} > 0$; (ii) (Symmetry) $\mathbf{W} = \mathbf{W}^\top$; (iii) (Null space property) null$\{\mathbf{I} - \mathbf{W}\} = $ span$\{\mathbf{1}\}$; (iv) (Spectral property) $\mathbf{I} \succeq \mathbf{W} \succ -\mathbf{I}$.* Under these properties, the eigenvalues of $\mathbf{W}$ satisfies $1 = \lambda_1(\mathbf{W})) > \lambda_2(\mathbf{W})) \geq \cdots \geq \lambda_m(\mathbf{W})) > -1$. And $\lambda := \max\{|\lambda_2(\mathbf{W})|, |\lambda_m(\mathbf{W}))|\}$ and $1 - \lambda \in (0,1]$ is the spectral gap of $\mathbf{W}$, which usually measures the degree of the network topology.

**Decentralized Partial Personalized Models.** Below, we present a general setting of DFL with *partial model personalization* for considering the communication overhead. Specifically, the model parameters are partitioned into two parts: the *shared* parameters $u \in \mathbb{R}^{d_0}$ and the *personal* parameters $v_i \in \mathbb{R}^{d_i}$ for $i = 1, \ldots, m$. The full model on client $i$ is denoted as $w_i = (u_i, v_i)$. To simplify presentation, we denote $V = (v_1, \ldots, v_m) \in \mathbb{R}^{d_1 + \cdots + d_m}$, and then our goal is to solve this problem:

$$\min_{u,V} \quad F(u,V) := \frac{1}{m} \sum_{i=1}^{m} F_i(u, v_i), \tag{2}$$

where $u$ denotes the consensus model averaged with all shared models $u_i$, that is $u = \frac{1}{m} \sum_{i=1}^{m} u_i$. Moreover, we consider the more general non-convex setting $F_i(u_i, v_i) = E_{\xi_i \sim \mathcal{D}_i}[F_i(u_i, v_i; \xi_i)]$ and use $\nabla_u$ and $\nabla_v$ to represent stochastic gradients with respect to $u_i$ and $v_i$, respectively.

In the DFL setting, the shared parameters $u_i$ of each client $i$ are sent out to the neighbors of client $i$ from the neighborhood set with adjacency matrix $\mathbf{W}$, which records the communication connections between any two clients (communication topology). In contrast, the personal parameters $v_i$ only perform multiple local iterations in each client $i$ and do not be sent out.

### 3.2   DFEDMDC AND DFEDSMDC ALGORITHMS

In this subsection, we first demonstrate the choice of the shared model parts and then present the DFedMDC and DFedSMDC algorithms for solving problem (2). The detailed procedure and pipeline are presented in Algorithm 1 and Figure 1, respectively.

**Partial Model Personalization.** Drawing from previous research on CNNs, layers that serve specific engineering purposes: lower convolution layers (close to the input) are responsible for feature extraction, and the upper linear layers (close to the output) focus on complex pattern recognition (Pillutla et al., 2022). The feature extraction layers, mapping data from high-dimensional feature space to an easily distinguished low space, are similar between clients but prone to over-fitting. The linear classification layers, which determine the data category from the output of the previous feature extraction layers, are very different from data heterogeneity clients (Li et al., 2023). Therefore, when

averaging the clients' full model will prevent over-fitting for feature extraction layers, but it may disturb local data classification in personalized tasks. Based on the substantial views, we set the feature extraction layers as the shared parts and the linear classification layers as the personalized parts. Similar choices can be seen in (Collins et al., 2021; Oh et al., 2021; Pillutla et al., 2022)

**DFedMDC.** We present DFedMDC to explore the possible partial personalization benefit of decentralized FL. It leverages the alternating update approach for model training to better fit the aggregated shared parts. Specifically, the personal parameters $v_i$ for each client perform multiple local iterations at first in line 6. While the mixing shared part represents the consensus information among clients, the personal part, only containing the local information is not compatible enough with the mixing shared one. This first step is to increase compatibility between the personalized and the shared parts. After that, the shared parameters $u_i$ perform multiple local iterations in line 10. After multiple local iterations of shared parameters $u_i$ in each client $i$, the resulting parameters $z_i^t \leftarrow u_i^{t,K_u}$ is sent to its neighbors in line 12. Then each client updates its shared parameters by averaging its neighbors' shared parameters (including itself).

---

**Algorithm 1:** DFedMDC and DFedSMDC

**Input** : Total number of devices $m$, total number of communication rounds $T$, local learning rate $\eta_u$ and $\eta_v$, total number of local iterates $K_u$ and $K_v$.

**Output :** Personalized model $u_i^T$ and $v_i^T$.

1 **Initialization:** Randomly initialize each device's shared parameters $u_i^0$ and personal parameters $v_i^0$.

2 **for** $t = 0$ **to** $T - 1$ **do**

3    **for** *client $i$ in parallel* **do**

4       Set $u_i^{t,0} \leftarrow u_i^t$ and sample a batch of local data $\xi_i$ and calculate local gradient iteration.

5       **for** $k = 0$ **to** $K_v - 1$ **do**

6          Perform personal parameters $v_i$ update:
$$v_i^{t,k+1} = v_i^{t,k} - \eta_v \nabla_v F_i(u_i^{t,0}, v_i^{t,k}; \xi_i).$$

7       **end**

8       $v_i^{t+1} \leftarrow v_i^{t,K_v}$.

9       **for** $k = 0$ **to** $K_u - 1$ **do**

10          Update shared parameters $u_i$ via Option I or II.

11       **end**

12       $z_i^t \leftarrow u_i^{t,K_u}$. Receive neighbors' shared models $z_j^t$ with adjacency matrix $\boldsymbol{W}$:
$$u_i^{t+1} = \sum_{l \in \mathcal{N}(i)} w_{i,l} z_i^t.$$

13    **end**

14 **end**

15 Option I: (DFedMDC) Find a minimum for $u_i$ with SGD

16 $u_i^{t,k+1} = u_i^{t,k} - \eta_u \nabla_u F_i(u_i^{t,k}, v_i^{t+1}; \xi_i).$

17 Option II: (DFedSMDC) Find a minimum for $u_i$ with SAM

18 $\epsilon(u_i^{t,k}) = \rho \frac{\nabla_u F_i(u_i^{t,k}, v_i^{t+1}; \xi_i)}{\|\nabla_u F_i(u_i^{t,k}, v_i^{t+1}; \xi_i)\|_2}.$

19 $u_i^{t,k+1} = u_i^{t,k} - \eta_u \nabla_u F_i(u_i^{t,k} + \epsilon(u_i^{t,k}), v_i^{t+1}; \xi_i).$

---

**An Enhanced Algorithm: DFedSMDC.** In FL, the model inconsistency issue is a major challenge across clients due to data heterogeneity (Shi et al., 2023b; Sun et al., 2022), resulting in severe over-fitting of local models. In particular, sparse communication topology is also a key factor in this issue in DFL (Shi et al., 2023b). Therefore, to further make partial personalization more suitable for DFL by decreasing the generalization error of shared parameters, we propose DFedSMDC, which integrates the SAM optimizer into the local iteration update of shared parameters $u_i$. Specifically, we adopt proper perturbation in the direction of the local gradient of the shared parameters $u_i$. At first, the gradient $\nabla_u F_i(u_i^{t,k}, v_i^{t+1}; \xi_i)$ of $u_i$ is calculated on mini-batch data $\xi_i$ for each client $i$. And then, we calculate the perturbation value in line 18, where $\rho$ is a hyper-parameter for controlling the value of the perturbation radius. Finally, adding the perturbation term into the direction of gradient $\nabla_u F_i(u_i^{t,k}, v_i^{t+1}; \xi_i)$ in line 18. The local averaging of $u_i$ is the same as the DFedMDC algorithm.

## 4 THEORETICAL ANALYSIS

In this section, we present the convergence analysis in DFedMDC and DFedSMDC methods for the characterization of convergence speed and the exploration of how partial personalization and SAM optimizer work. Below, we state some general assumptions at first (Pillutla et al., 2022).

**Assumption 1** (Smoothness). *For each client $i = \{1, \ldots, m\}$, the function $F_i$ is continuously differentiable. There exist constants $L_u, L_v, L_{uv}, L_{vu}$ such that for each client $i = \{1, \ldots, m\}$:*

- $\nabla_u F_i(u_i, v_i)$ *is $L_u$–Lipschitz with respect to $u_i$ and $L_{uv}$–Lipschitz with respect to $v_i$;*

- $\nabla_v F_i(u_i, v_i)$ *is $L_v$–Lipschitz with respect to $v_i$ and $L_{vu}$–Lipschitz with respect to $u_i$.*

*We summarize the relative cross-sensitivity of $\nabla_u F_i$ with respect to $v_i$ and $\nabla_v F_i$ with respect to $u$ with the scalar*

$$\chi := \max\{L_{uv}, L_{vu}\} / \sqrt{L_u L_v}.$$

**Assumption 2** (Bounded Variance). *The stochastic gradients in Algorithm 1 have bounded variance. That is, for all $u_i$ and $v_i$, there exist constants $\sigma_u$ and $\sigma_v$ such that*

$$\mathbb{E}\big[\big\|\nabla_u F_i(u_i, v_i; \xi_i) - \nabla_u F_i(u_i, v_i)\big\|^2\big] \le \sigma_u^2, \ \mathbb{E}\big[\big\|\nabla_v F_i(u_i, v_i; \xi_i) - \nabla_v F_i(u_i, v_i)\big\|^2\big] \le \sigma_v^2.$$

**Assumption 3** (Partial Gradient Diversity). *There exist a constant $\delta \ge 0$ such that*

$$\tfrac{1}{m}\textstyle\sum_{i=1}^m \big\|\nabla_u F_i(u_i, v_i) - \nabla_u F(u_i, V)\big\|^2 \le \delta^2, \ \forall u_i, \ V.$$

The above assumptions are mild and commonly used in the convergence analysis of FL (Sun et al., 2022; Shi et al., 2023b; Yang et al., 2021; Bottou et al., 2018; Reddi et al., 2021; Qu et al., 2022).

**About the Challenges of Convergence Analysis.** Due to the central server being discarded, various communication connections will become an important factor for decentralized optimization. Furthermore, communication is more careful in general classical FL scenarios rather than computation (McMahan et al., 2017; Li et al., 2020b; Kairouz et al., 2021; Qu et al., 2022). So the client adopts multi-step local iterations such as FedAvg (McMahan et al., 2017), which may lead to the local gradient failing to be unbiased. Because of these factors, technical difficulty exists in our theoretical analysis. How to analyze the convergence of decomposed model parameters while delivering the impact of communication topology. In this paper, we adopt the averaged shared parameter $\bar{u}^t = \frac{1}{m}\sum_{i=1}^m u_i^t$ of all clients to be the approximated solution of problem (2) due to only the shared parameters being communicated with the neighbors (Sun et al., 2022; Shi et al., 2023b). Now, we present the rigorous convergence rate of DFedMDC and DFedSMDC algorithms as follows.

**Theorem 1** (Convergence Analysis for DFedMDC). *Under assumptions 1-3 and definition 1, the local learning rates satisfy $\eta_u = \mathcal{O}(1/L_u K_u \sqrt{T}), \eta_v = \mathcal{O}(1/L_v K_v \sqrt{T})$, $F^*$ is denoted as the minimal value of $F$, i.e., $F(\bar{u}, V) \ge F^*$ for all $\bar{u} \in \mathbb{R}^d$, and $V = (v_1, \dots, v_m) \in \mathbb{R}^{d_1 + \dots + d_m}$. Let $\bar{u}^t = \frac{1}{m}\sum_{i=1}^m u_i^t$ and denote $\Delta_{\bar{u}}^t$ and $\Delta_v^t$ as:*

$$\Delta_{\bar{u}}^t = \big\|\nabla_u F(\bar{u}^t, V^t)\big\|^2, \quad and \quad \Delta_v^t = \tfrac{1}{m}\textstyle\sum_{i=1}^m \big\|\nabla_v F_i(u_i^t, v_i^t)\big\|^2.$$

*Therefore, we have the convergence rate as below:*

$$\frac{1}{T}\sum_{i=1}^T \Big(\frac{1}{L_u}\mathbb{E}[\Delta_{\bar{u}}^t] + \frac{1}{L_v}\mathbb{E}[\Delta_v^t]\Big) \le \mathcal{O}\Big(\frac{F(\bar{u}^1, V^1) - F^*}{\sqrt{T}} + \frac{\sigma_1^2}{(1-\lambda)^2\sqrt{T}} + \frac{\sigma_2^2}{\sqrt{T}} + \frac{\sigma_3^2}{(1-\lambda)^2 T}\Big). \quad (3)$$

*where*

$$\sigma_1^2 = \frac{\chi^2 L_v(\sigma_u^2 + \delta^2)}{L_u} + \frac{\sigma_u^2 + \delta^2}{L_u^2}, \quad \sigma_2^2 = \frac{\sigma_v^2(L_v + 1)}{L_v^2} + \frac{\sigma_u^2 + \delta^2}{K_u L_u}, \quad \sigma_3^2 = \frac{\sigma_u^2 + \delta^2}{K_u L_u}.$$

**Remark 1.** These variables have a significant influence on the convergence bound. Specifically, measuring the statistical heterogeneity, such as local variance $\sigma_u^2, \sigma_v^2$ and global diversity, the smoothness of local loss functions such as $L_u, L_v$, and $L_{vu}$, and the communication topology measured by $1 - \lambda$. There does not exist $L_{uv}$ due to the alternate update ($v_i$ first, then $u$). More details are in Appendix C.

**Theorem 2** (Convergence Analysis for DFedSMDC). *Under assumptions 1-3 and definition 1, the local learning rates satisfy $\eta_u = \mathcal{O}(1/L_u K_u \sqrt{T}), \eta_v = \mathcal{O}(1/L_v K_v \sqrt{T})$. Let $\bar{u}^t = \frac{1}{m}\sum_{i=1}^m u_i^t$ and denote $\Delta_{\bar{u}}^t$ and $\Delta_v^t$ as Theorem 1. When the perturbation amplitude $\rho$ is proportional to the learning rate, e.g., $\rho = \mathcal{O}(1/\sqrt{T})$, the sequence of outputs $\Delta_{\bar{u}}^t$ and $\Delta_v^t$ generated by DFedSMDC, we have:*

$$\frac{1}{T}\sum_{i=1}^T \Big(\frac{1}{L_u}\mathbb{E}[\Delta_{\bar{u}}^t] + \frac{1}{L_v}\mathbb{E}[\Delta_v^t]\Big) \le \mathcal{O}\Big(\frac{F(\bar{u}^1, V^1) - F^*}{\sqrt{T}} + \frac{\sigma^2 L_{vu}^2}{(1-\lambda)^2\sqrt{T}} + \frac{\sigma_4^2}{\sqrt{T}} + \frac{\sigma^2 L_u}{(1-\lambda)^2 T}\Big). \quad (4)$$

*where $\sigma_4^2 = \frac{\sigma_v^2(L_v+1)}{L_v^2} + \frac{L_u^2\rho^2 + \sigma_u^2 + \delta^2}{K_u L_u}$ and $\mathcal{O}(\sigma^2) = \mathcal{O}\Big(\frac{\rho^2}{K_u} + \frac{\sigma_u^2 + \delta^2}{L_u^2}\Big) = \mathcal{O}\Big(\frac{1}{K_u T} + \frac{\sigma_u^2 + \delta^2}{L_u^2}\Big).$*

**Remark 2.** It is clear that the bound is facilitated via SAM optimizer from the smoothness-enabled perspective, such as $L_u^2$ and $L_{vu}^2$. Thus, the shared model $u_i$ may be flatter, thereby decreasing the generalization error of the whole model $w_i = (u_i, v_i)$. Finally, the shared parameters $u_i$ aggregation process is promoted, thereby achieving better performance.

Table 2: Test accuracy (%) on CIFAR-10 & 100 in both Dirichlet and Pathological distribution settings.

| Algorithm | CIFAR-10 | | | | CIFAR-100 | | | |
|---|---|---|---|---|---|---|---|---|
| | Dirichlet | | Pathological | | Dirichlet | | Pathological | |
| | $\alpha = 0.1$ | $\alpha = 0.3$ | c = 2 | c = 5 | $\alpha = 0.1$ | $\alpha = 0.3$ | c = 5 | c = 10 |
| Local | $78.96_{\pm.42}$ | $63.20_{\pm.28}$ | $85.16_{\pm.18}$ | $68.56_{\pm.35}$ | $39.38_{\pm.33}$ | $22.59_{\pm.49}$ | $71.34_{\pm.46}$ | $53.15_{\pm.31}$ |
| FedAvg | $84.17_{\pm.28}$ | $79.66_{\pm.20}$ | $85.04_{\pm.11}$ | $82.80_{\pm.27}$ | $57.43_{\pm.03}$ | $57.01_{\pm.06}$ | $69.05_{\pm.43}$ | $66.37_{\pm.48}$ |
| FedSAM | $84.17_{\pm.75}$ | $80.02_{\pm.27}$ | $84.99_{\pm.04}$ | $81.18_{\pm.21}$ | $57.35_{\pm.38}$ | $55.12_{\pm.67}$ | $69.29_{\pm.85}$ | $66.10_{\pm.35}$ |
| FedPer | $88.57_{\pm.09}$ | $84.06_{\pm.29}$ | $90.94_{\pm.24}$ | $86.97_{\pm.35}$ | $54.23_{\pm.14}$ | $34.07_{\pm.76}$ | $78.48_{\pm.93}$ | $70.38_{\pm.02}$ |
| FedRep | $88.78_{\pm.40}$ | $84.50_{\pm.05}$ | $91.09_{\pm.12}$ | $86.22_{\pm.51}$ | $44.02_{\pm.98}$ | $26.88_{\pm.49}$ | $78.77_{\pm.19}$ | $68.15_{\pm.43}$ |
| FedBABU | $87.79_{\pm.53}$ | $83.26_{\pm.09}$ | $91.32_{\pm.15}$ | $84.90_{\pm.24}$ | $60.23_{\pm.07}$ | $52.37_{\pm.82}$ | $77.50_{\pm.33}$ | $69.81_{\pm.12}$ |
| Fed-RoD | $89.15_{\pm.12}$ | $85.68_{\pm.08}$ | $90.10_{\pm.04}$ | $87.81_{\pm.45}$ | $65.79_{\pm.05}$ | $58.54_{\pm.69}$ | $80.50_{\pm.45}$ | $73.59_{\pm.15}$ |
| Ditto | $80.22_{\pm.10}$ | $73.51_{\pm.04}$ | $84.96_{\pm.40}$ | $75.59_{\pm.32}$ | $48.85_{\pm.54}$ | $48.65_{\pm.50}$ | $69.48_{\pm.45}$ | $60.77_{\pm.30}$ |
| DFedAvgM | $87.39_{\pm.13}$ | $82.60_{\pm.18}$ | $90.72_{\pm.08}$ | $84.69_{\pm.25}$ | $59.76_{\pm.69}$ | $54.98_{\pm.48}$ | $76.70_{\pm.59}$ | $71.08_{\pm.52}$ |
| Dis-PFL | $87.77_{\pm.46}$ | $82.71_{\pm.28}$ | $88.19_{\pm.47}$ | $82.29_{\pm.61}$ | $56.06_{\pm.20}$ | $46.65_{\pm.18}$ | $71.79_{\pm.42}$ | $65.35_{\pm.10}$ |
| DFedSAM | $84.96_{\pm.30}$ | $77.36_{\pm.11}$ | $90.14_{\pm.22}$ | $83.05_{\pm.40}$ | $58.21_{\pm.53}$ | $47.80_{\pm.49}$ | $74.25_{\pm.17}$ | $67.34_{\pm.43}$ |
| DFedMDC | $88.85_{\pm.21}$ | $86.50_{\pm.05}$ | $91.26_{\pm.23}$ | $86.85_{\pm.37}$ | $66.26_{\pm.25}$ | $57.66_{\pm.42}$ | $78.78_{\pm.41}$ | $72.19_{\pm.21}$ |
| DFedSMDC | $\mathbf{91.08}_{\pm.34}$ | $\mathbf{87.67}_{\pm.22}$ | $\mathbf{92.20}_{\pm.14}$ | $\mathbf{88.34}_{\pm.31}$ | $\mathbf{67.03}_{\pm.36}$ | $\mathbf{58.73}_{\pm.19}$ | $\mathbf{80.82}_{\pm.33}$ | $\mathbf{74.50}_{\pm.35}$ |

**Remark 3.** In both Theorems 1&2, the convergence bounds are related to the spectral gap $(1 - \lambda)$ of the communication topology, which is associated with the participation clients. From the relationship between the spectral gap and the participation of clients in Table 1 we can see that the convergence bounds of different topologies are ranked as Fully-connected > Exponential > Grid > Ring.

**Remark 4.** Compared with the SOTA bounds $\mathcal{O}\left(\frac{1}{\sqrt{T}} + \frac{\sigma_l^2 + K\sigma_g^2}{K\sqrt{T}} + \frac{\sigma_l^2 + K\sigma_g^2 + KB^2}{K(1-\lambda)^2 T^{3/2}}\right)$ of DFedAvg (Sun et al., 2022) and $\mathcal{O}\left(\frac{1}{\sqrt{KT}} + \frac{K(\sigma_g^2 + \sigma_l^2)}{T} + \frac{\sigma_g^2 + \sigma_l^2}{K^{1/2}(1-\lambda)^2 T^{3/2}}\right)$ of DFedSAM (Shi et al., 2023b) in decentralized works, our algorithms reflect the impact of the L-smoothness and the gradient variance of the shared model $u$ and personalized model $v$ on convergence rate. On the other hand, compared with the SOTA bound of FedAlt (Pillutla et al., 2022) in PFL, our algorithms reflect the impact of the communication topology $(1 - \lambda)$ (the value of that increases when connectivity is more sparse).

## 5 EXPERIMENTS

### 5.1 EXPERIMENT SETUP

**Dataset and Data Partition.** We evaluate our approaches on CIFAR-10, CIFAR-100 (Krizhevsky et al., 2009), and Tiny-ImageNet (Le and Yang, 2015) datasets with Dirichlet and Pathological data partition. All detailed experiments on the Tiny-ImageNet dataset are placed in **Appendix** B.5 due to the limited space. We partition the training and testing data according to the same Dirichlet distribution Dir($\alpha$) such as $\alpha = 0.1$ and $\alpha = 0.3$ for each client. The smaller the $\alpha$ is, the more heterogeneous the setting is. Meanwhile, for each client, we sample 2 and 5 classes from a total of 10 classes on CIFAR-10, and 5 and 10 classes from a total of 100 classes on CIFAR-100 respectively. The number of sampling classes is represented as "c" in Table 2 and the fewer classes each client owns, the more heterogeneous the setting is.

Table 1: Spectral Gap $1 - \lambda$ of communication topologies.

| Graph Topology | Spectral Gap $1 - \lambda$ |
|---|---|
| Fully-connected | 1 |
| Disconnected | 0 |
| Ring | $\approx 16\pi^2/3m^2$ |
| Grid | $\mathcal{O}(1/(m\log_2(m)))$ |
| EXponential | $2/(1 + \log_2(m))$ |

**Baselines and Backbone.** We compare the proposed methods with the SOTA baselines PFL. For instance, Local is the simplest method of conducting training on their own data without communicating with other clients. CFL methods include FedAvg (McMahan et al., 2017), FedSAM (Foret et al., 2020), FedPer (Arivazhagan et al., 2019), FedRep (Collins et al., 2021), FedBABU (Oh et al., 2021), Fed-RoD (Chen and Chao, 2021) and Ditto (Li et al., 2021). For DFL methods, we take DFedAvgM (Sun et al., 2022), Dis-PFL (Dai et al., 2022) as our baselines. All methods are evaluated on ResNet-18 (He et al., 2016) and replace the batch normalization with the group normalization followed by (Dai et al., 2022) to avoid unstable performance.

**Implementation Details.** We perform 500 rounds with 100 clients on CIFAR-10 & CIFAR-100. The client sampling radio is 0.1 in CFL, while each client communicates with 10 neighbors in PFL accordingly. The batch size is 128. We set SGD (Robbins and Monro, 1951) as the base optimizer with a learning rate of 0.1 and local momentum of 0.9. We report the mean performance with 3 different seeds. More hyperparameter details can be found in **Appendix** B.2& B.3.

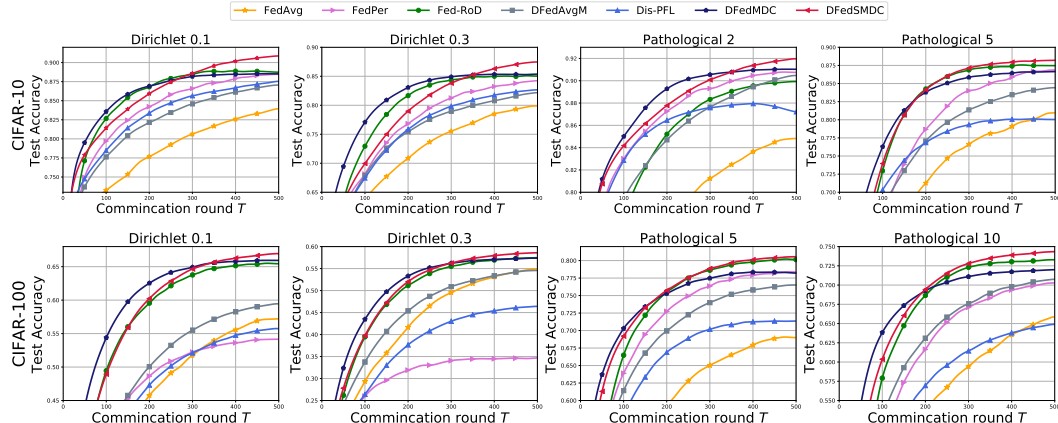

Figure 2: Test accuracy on CIFAR-10 (first line) and CIFAR-100 (second line) with heterogenous data partitions. With limited pages, we only show the training progress of the typical methods.

## 5.2 PERFORMANCE EVALUATION

**Comparison with the baselines.** As shown in Table 2 and Figure 2, the proposed DFedMDC and DFedSMDC outperform other baseline methods with the best stability and better performance in both different dataset and different data heterogeneity scenarios. Specifically, on the CIFAR-10 dataset, DFedMDC and DFedSMDC achieve 87.67% and 86.50% on the Directlet-0.3 setups, 0.82% and 1.99% ahead of the best comparing method Fed-RoD. On the CIFAR-100 dataset, DFedSMDC achieves at least 0.32% and 0.91% improvement from the other baselines on the Pathological-5 and Pathological-10 settings. We attribute thate DFedMDC and DFedSMDC learn the "right" part with their neighbors and allow the private linear classifier to better adapt to their local data. It is also observed that DFedSMDC outperform DFedMDC stability, demonstrating that the SAM optimizer significantly improve the shared feature extractor.

**Hyperparameters Sensitivity.** We discuss two data heterogeneity of Dirichlet distribution and Pathological distribution in Table 2, and prove the effectiveness and robustness of the proposed methods. In Dirichlet distribution, since the local training can't cater for all classes inside clients, the accuracy decreases with the level of heterogeneity decreasing. On CIFAR-10, when the heterogeneity decreases from 0.1 to 0.3, Fed-RoD drops from 89.15% to 85.68%, while DFedSMDC drops about 3.41% to 87.67%, meaning its stronger stability for several heterogeneous settings. Pathological distribution defines limited classes for each client which is a higher level of heterogeneity. DFedSMDC is 0.88% ahead of the best compared CFL method on CIFAR-10 with only 2 categories per client and 0.91% ahead on CIFAR-100 dataset with only 10 categories per client. The results confirm that the proposed methods could achieve better performance with the strong heterogeneity.

Table 3: The required communication rounds when achieving the target accuracy (%).

| Algorithm | CIFAR-10 | | CIFAR-100 | |
|---|---|---|---|---|
| | Dir-0.3 | Pat-2 | Dir-0.3 | Pat-10 |
| | acc@80 | acc@90 | acc@45 | acc@65 |
| FedAvg | - | - | 234 | 456 |
| FedSAM | - | - | 221 | 416 |
| FedPer | 262 | 343 | - | 246 |
| FedRep | 189 | 322 | - | 225 |
| FedBABU | 270 | 312 | 261 | 314 |
| Fed-RoD | 170 | 462 | 133 | 148 |
| Ditto | - | - | 279 | - |
| DFedAvgM | 354 | 439 | 192 | 230 |
| Dis-PFL | 307 | - | 368 | 492 |
| DFedSAM | - | 465 | 367 | 344 |
| DFedMDC | **131** | **224** | **111** | **113** |
| DFedSMDC | 160 | 280 | 131 | 139 |

**Convergence speed.** We illustrate the convergence speed via the learning curves of the compared methods in Figure 2 and collect the communication rounds to reach a target accuracy (acc@) in Table 3. DFedMDC achieves the fastest convergence speed among the comparison methods, which benefits from the direct partial model exchange and alternate update. In comparison with the CFL methods, directly learning the neighbors' feature representation in DFL can speed up the convergence rate for personalized problems. Also, the difference between DFedMDC and DFedAvgM indicates that the convergence speed of alternate updating is faster than uniform updating. Notably, we target the setting where the busiest node's communication bandwidth is restricted for fairness when compared with the CFL methods.

**Impact of communication topologies.** For decentralized methods, the performance under various communication topologies will help to evaluate the robustness of the methods. Combining the topology visualization in **Appendix** B.4 and the spectral Gap analysis in Table 1, both the convergence bounds and communication costs of different topologies are ranked as Fully-connected > Exponential > Grid > Ring. In figure 3, from sparse connection to compact connection(Ring-Grid-Exp-Full), all the methods achieve better performance. Besides, DFedMDC and DFedSMDC are more robust in various communication topologies.

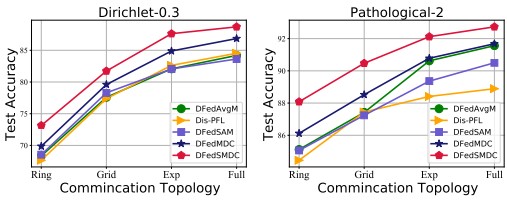

Figure 3: Personal test accuracy (%) in various network topologies for the DFL methods on CIFAR-10.

## 5.3 ABLATION STUDY

**Integrating SAM into the shared model or personal model or whole model.** We investigate the effect of adding the SAM optimizer to different parts with different data heterogeneity on the CIFAR-10 dataset. From Table 4, DFedSMDC-U (SAM only for the shared model, dubbed as "body") achieves the best in the Dirichlet setting and DFedSMDC-UV (adding SAM to both shared

Table 4: Test accuracy (%) of different model parts with the SAM optimizer.

| Algorithm | Body | Head | Dirichlet | Pathological |
|---|---|---|---|---|
| DFedMDC | | | 86.50 | 91.26 |
| DFedSMDC-U | ✓ | | **87.67** | 92.20 |
| DFedSMDC-V | | ✓ | 86.46 | 91.50 |
| DFedSMDC-UV | ✓ | ✓ | 87.43 | **92.58** |

and personal parts) achieves the best in the Pathological setting. From the difference between DFed-MDC, DFedSMDC-U and DFedSMDC-UV, we observe that the SAM optimizer, uniformly reducing the inconsistency of the feature extractor among clients, can improve the feature extraction ability of the shared parts. Besides, the comparison from DFedMDC, DFedSMDC-V (SAM only for the personal model, dubbed as "head") and DFedSMDC-UV illustrates that the benefits of adding SAM to the personal model are sensitive to the data distribution and hyperparameter setup. Thus, we set DFedSMDC-U as our default algorithm and denote it as DFedSMDC in the main experiments.

**Effectiveness of local epochs.** In Figure 4, we illustrate the effect of local epochs for the personal parameters in different heterogeneity scenarios on the CIFAR-10 after 200 communication rounds. With fixed local epochs of 5 for the shared parameters, more local epochs for the personal parts helps to learn a more effective personal model in the Dirichlet scenarios. In the Pathological scenarios, the local epochs for the personal parameters need a trade-off to improve the shared extraction ability and adapt the personal local data.

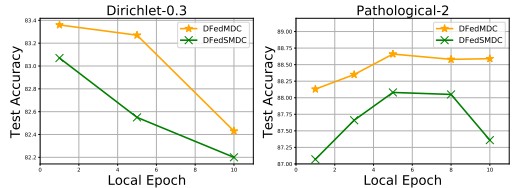

Figure 4: Effect of the local epochs for the personal parameters $v_i$ in client $i$.

**Number of participated clients.** We compare the personalized performance between different numbers of client participation on the CIFAR-10 dataset with Dirichlet-0.3 in Figure 5. Compared with larger participated clients {50, 100, 200}, the smaller participated clients {5, 10} can achieve better test accuracy and convergence as the number of local data increases.

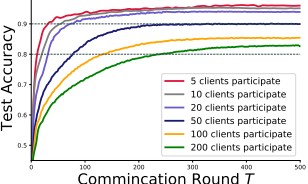

Figure 5: Effect of the clients' size.

## 6 CONCLUSION

In this paper, we propose novel methods DFedMDC and DFedSMDC for PFL, which simultaneously guarantee robust communication and better model performance with convergence guarantee via adopting decentralized partial model personalization based on model decoupling. It efficiently personalizes the "right" components in the deep modern models and alternatively updates the shared parameters and personal parameters in a peer-to-peer manner. For theoretical findings, we present the convergence rate in the stochastic non-convex setting for DFedMDC and DFedSMDC. Empirical results also verify the superiority of our approaches.

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

# Supplementary Material for " Enhancing Personal Decentralized Federated Learning through Model Decoupling "

In this part, we provide the supplementary materials including more introduction to the related works, experimental details and results, and the proof of the main theorem.

- **Appendix** A: More details in the related works.
- **Appendix** B: More details in the experiments.
- **Appendix** C: Proof of the theoretical analysis.

## A  MORE DETAILS IN THE RELATED WORKS

**Decentralized/Distributed Training.** By combining SGD and gossip, early work achieved decentralized training and convergence of the model in (Blot et al., 2016). D-PSGD (Lian et al., 2017) is the classic decentralized parallel SGD method. FastMix (Ye et al., 2020) investigates the advantage of increasing the frequency of local communications within a network topology, in which the optimal computational complexity and near-optimal communication complexity are established. DeEPCA (Ye and Zhang, 2021) integrates FastMix into a decentralized PCA algorithm to accelerate the training process. DeLi-CoCo (Hashemi et al., 2022) performs multiple compression gossip steps in each iteration for fast convergence with arbitrary communication compression. Network-DANE (Li et al., 2020a) uses multiple gossip steps and generalizes DANE to decentralized scenarios. The work in (Lin et al., 2021) modifies the momentum term of decentralized SGD (DSGD) to be adaptive to heterogeneous data, while the work in (Hsieh et al., 2020) replaces batch norm with layer norm. (Li et al., 2022b) dynamically updates the mixing weights based on meta-learning and learns a sparse topology to reduce communication costs. The work in (Zhu et al., 2022) provides the topology-aware generalization analysis for DSGD, they explore the impact of various communication topologies on the generalizability.

**Sharpness Aware Minimization (SAM).** SAM (Qu et al., 2022) is an effective optimizer for training deep learning (DL) models, which leverages the flatness geometry of the loss landscape to improve model generalization ability. Recently, the work in (Andriushchenko and Flammarion, 2022) studies the properties of SAM and provides convergence results of SAM for non-convex objectives. As a powerful optimizer, SAM and its variants have been applied to various DL tasks (Zhao et al., 2022; Kwon et al., 2021; Du et al., 2021; Liu et al., 2022; Abbas et al., 2022; Mi et al., 2022; Zhong et al., 2022; Huang et al.) and FL tasks (Qu et al., 2022; Caldarola et al., 2022; Sun et al.; 2023; Shi et al., 2023b;c;a). For instance, the works in (Qu et al., 2022), (Sun et al.), (Zhu et al., 2023) and (Caldarola et al., 2022) integrate SAM to improve the generalization, and thus mitigate the distribution shift and achieve a new SOTA performance for FL.

## B  MORE DETAILS IN THE EXPERIMENT

In this section, we provide more details of our experiments and more extensive experimental results to compare the performance of the proposed DFedMDC and DFedSMDC against other baselines.

### B.1  DATASETS AND DATA PARTITION

Table 5: The details on the CIFAR-10 and CIFAR-100 datasets.

| Dataset | Training Data | Test Data | Class | Size |
|---|---|---|---|---|
| CIFAR-10 | 50,000 | 10,000 | 10 | 3×32×32 |
| CIFAR-100 | 50,000 | 10,000 | 100 | 3×32×32 |
| Tiny-ImageNet | 100,000 | 10,000 | 200 | 3×64×64 |

CIFAR-10/100 and Tiny-ImageNet are three basic datasets in the computer version study. As shown in Table 5, they are all colorful images with different classes and different resolutions. We use two non-IID partition methods to split the training data in our implementation. One is based on Dirichlet distribution on the label ratios to ensure data heterogeneity among clients, where a smaller $\alpha$ means higher heterogeneity. Another assigns each client a limited number of categories, called Pathological distribution, where fewer categories mean higher heterogeneity. The distribution of the test datasets is the same as in training datasets. We run 500 communication rounds for CIFAR-10, CIFAR-100, and 300 rounds for Tiny-ImageNet.

## B.2 MORE DETAILS ABOUT THE DFEDMDC AND DFEDSMDC

For DFedMDC and DFedSMDC, we train the shared part for 5 epochs per round as the same as other baselines; for the personal part, we find that training 1 epoch and 5 epochs is the most appropriate parameter on Dirichlet distribution and Pathological distribution respectively. Therefore, we conduct ablation experiments to select the most appropriate parameters for each data distribution in Figure 4. We set SGD as the base optimizer with a learning rate $\eta_v = 0.001$ for the personal and $\eta_u = 0.1$ for shared parameters update with a decay rate of 0.005 and local momentum of 0.9. The weight perturbation ratio in DFedSMDC is set to $\rho = 0.7$.

## B.3 MORE DETAILS ABOUT BASELINES

**Local** is the simplest method for personalized learning. It only trains the personalized model on the local data and does not communicate with other clients. For the fair competition, we train 5 epochs locally in each round.

**FedAvg** (McMahan et al., 2017) is the most commonly discussed method in FL. It selects some clients to perform local training on each dataset and then aggregates the trained local models to update the global model. Actually, the local model in FedAvg is also the comparable personalized model for each client.

**FedSAM** (Foret et al., 2020) leverages gradient perturbation to generate local flat models via Sharpness Aware Minimization (SAM). The communication and client selection are the same as in FedAvg. We set the perturbation radius $\rho = 0.7$ in our experiments.

**FedPer** (Arivazhagan et al., 2019) proposes a base + personalized layer approach for PFL to combat the ill effects of statistical heterogeneity. We set the linear layer as the personalized layer and the rest model as the base layer. It follows FedAvg's training paradigm but only passes the base layer to the server and keeps the personalized layer locally.

**FedRep** (Collins et al., 2021) also proposes a body(base layer) + head(personalized layer) framework like FedPer, but it fixes one part when updating the other. We follow the official implementation[2] to train the head for 10 epochs with the body fixed, and then train the body for 5 epochs with the head fixed.

**FedBABU** (Oh et al., 2021) is also a model split method that achieves good personalization via fine-tuning from a good shared representation base layer. Different from FedPer and FedRep, FedBABU only updates the base layer with the personalized layer fixed and finally fine-tunes the whole model. Following the official implementation[3], it fine-tunes 5 times in our experiments.

**Fed-RoD** (Chen and Chao, 2021) explicitly decouples a model's dual duties with two prediction tasks—generic optimization and personalized optimization and utilizes a hyper network to connect the generic model and the personalized model. Each client first updates the generic model with balanced risk minimization then updates the personalized model with empirical risk minimization.

**Ditto** (Li et al., 2021) achieves personalization via a trade-off between the global model and local objectives. It totally trains two models on the local datasets, one for the global model (similarly aggregated as in FedAvg) with local empirical risk, one for the personal model (kept locally) with both empirical and proximal terms towards the global model. We set the regularization parameters $\lambda$ as 0.75.

---

[2] https://github.com/lgcollins/FedRep
[3] https://github.com/jhoon-oh/FedBABU

**DFedAvgM** (Sun et al., 2022) is the decentralized FedAvg with momentum, in which clients only connect with their neighbors by an undirected graph. For each client, it first initials the local model with the received models then updates it on the local datasets with a local stochastic gradient. We choose $\rho = 0.7$ in our experiments.

**Dis-PFL** (Dai et al., 2022) employs personalized sparse masks to customize sparse local models in the PFL setting. Each client first initials the local model with the personalized sparse masks and updates it with empirical risk. Then filter out the parameter weights that have little influence on the gradient through cosine annealing pruning to obtain a new mask. Following the official implementation[4], the sparsity of the local model is set to 0.5 for all clients.

**DFedSAM** Shi et al. (2023b) leverages gradient perturbation to generate local flat models via Sharpness Aware Minimization (SAM). The communication framework between neighbors is the same as DFedAvgM, but the local update is performed by the SAM optimizer. We set the perturbation radius $\rho = 0.7$ in our experiments.

### B.4 COMMUNICATION NETWORK TOPOLOGIES

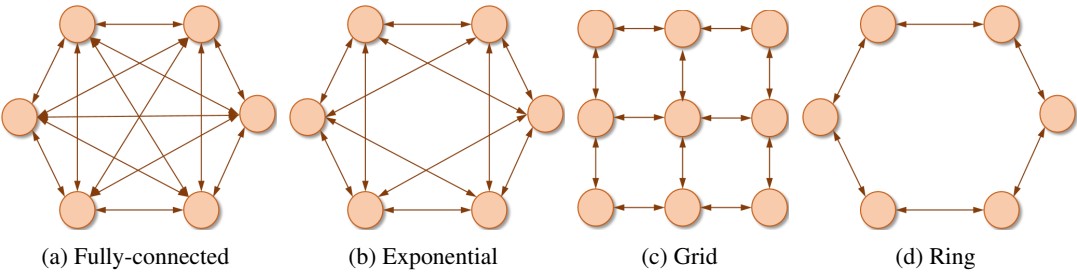

(a) Fully-connected     (b) Exponential     (c) Grid     (d) Ring

Figure 6: Illustration of the communication network topologies.

### B.5 MORE EXPERIMENTS RESULTS ON TINY IMAGENET

**Comparison with the baselines.** In Table 6 and Figure 7, we compare DFedMDC and DFedSMDC with other baselines on the Tiny-ImageNet with different data distributions. The comparison shows that the proposed methods have a competitive performance, especially under higher heterogeneity, e.g. for Dirichlet-0.1 and Pathological-10. Specifically in the Dirichlet-0.1 setting, DFedSMDC achieves 29.70%, at least 0.67% improvement from the CFL methods, while DFedSMDC and DFedMDC are 1.15% and 2.78% ahead of the other baselines in Pathological-10 setting. The original intention of our design is to build a great personalized model by focusing on local training and exchanging the feature extraction capabilities with neighbors via decentralized partial model training. So when the heterogeneity increases, our algorithms have a significant improvement.

---

[4] https://github.com/rong-dai/DisPFL

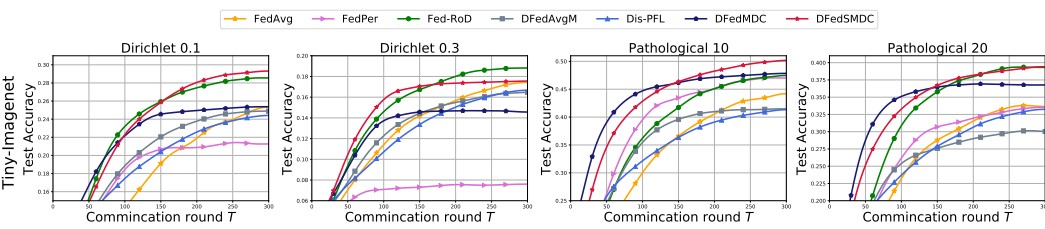

Figure 7: Test accuracy on Tiny-ImageNet with heterogenous data partitions. With limited pages, we only show the training progress of the typical methods.

Table 6: Test accuracy (%) on Tiny-ImageNet in both Dirichlet and Pathological distribution settings on Tiny-ImageNet.

| Algorithm | Tiny-ImageNet | | | |
| | Dirichlet | | Pathological | |
| | $\alpha = 0.1$ | $\alpha = 0.3$ | c = 10 | c = 20 |
| --- | --- | --- | --- | --- |
| Local | $12.13_{\pm.13}$ | $5.42_{\pm.21}$ | $28.49_{\pm.16}$ | $16.72_{\pm.34}$ |
| FedAvg | $25.55_{\pm.02}$ | $17.58_{\pm.25}$ | $44.56_{\pm.39}$ | $34.10_{\pm.59}$ |
| FedSAM | $26.32_{\pm.38}$ | $17.38_{\pm.19}$ | $41.85_{\pm.58}$ | $34.56_{\pm.49}$ |
| FedPer | $21.64_{\pm.72}$ | $7.71_{\pm.08}$ | $47.35_{\pm.03}$ | $33.68_{\pm.33}$ |
| FedRep | $17.54_{\pm.79}$ | $5.78_{\pm.05}$ | $46.76_{\pm.73}$ | $31.15_{\pm.54}$ |
| FedBABU | $27.40_{\pm.08}$ | $\mathbf{19.73}_{\pm.06}$ | $46.53_{\pm.20}$ | $38.68_{\pm.31}$ |
| Fed-RoD | $29.03_{\pm.55}$ | $19.25_{\pm.45}$ | $48.01_{\pm.40}$ | $39.28_{\pm.58}$ |
| Ditto | $21.71_{\pm.66}$ | $14.47_{\pm.14}$ | $40.65_{\pm.15}$ | $28.74_{\pm.38}$ |
| DFedAvgM | $25.29_{\pm.26}$ | $17.07_{\pm.17}$ | $42.80_{\pm.43}$ | $30.58_{\pm.51}$ |
| Dis-PFL | $24.71_{\pm.18}$ | $16.94_{\pm.36}$ | $41.93_{\pm.12}$ | $33.57_{\pm.62}$ |
| DFedSAM | $24.18_{\pm.32}$ | $16.92_{\pm.19}$ | $42.87_{\pm.31}$ | $32.61_{\pm.14}$ |
| DFedMDC | $25.71_{\pm.20}$ | $14.94_{\pm.44}$ | $49.16_{\pm.19}$ | $37.25_{\pm.27}$ |
| DFedSMDC | $\mathbf{29.70}_{\pm.47}$ | $17.81_{\pm.35}$ | $\mathbf{50.79}_{\pm.28}$ | $\mathbf{39.44}_{\pm.40}$ |

**Convergence speed.** We show the convergence speed of DFedMDC and DFedSMDC in Table 7 by reporting the number of rounds required to achieve the target personalized accuracy (acc@) on Tiny-ImageNet. We set the algorithm that takes the most rounds to reach the target accuracy as "1.00×", and find that the proposed DFedMDC and DFedSMDC achieve the fastest convergence speed on average (3.51× and 3.04× on average) among the SOTA PFL algorithms. Local training in PFL consistently pursues empirical risk minimization on the local datasets, which can efficiently train the personalized model fitting the local distribution. Also, the alternate updating mode will bring a comparable gain to the convergence speed from the difference between DFedMDC and DFedAvgM. Thus, our methods can efficiently train the personalized model, especially on the higher heterogeneity.

Table 7: The required communication rounds when achieving the target accuracy (%) on Tiny-ImageNet.

| Algorithm | Tiny-ImageNet | | | | | | | |
| | Dirichlet-0.1 | | Dirichlet-0.3 | | Pathological-10 | | Pathological-20 | |
| | acc@20 | speedup | acc@14 | speedup | acc@40 | speedup | acc@30 | speedup |
| --- | --- | --- | --- | --- | --- | --- | --- | --- |
| FedAvg | 160 | 1.11 × | 144 | 1.47 × | 192 | 1.36 × | 172 | 1.50 × |
| FedSAM | 136 | 1.31 × | 149 | 1.42 × | 214 | 1.22 × | 180 | 1.43 × |
| FedPer | 123 | 1.45 × | - | - | 103 | 2.53 × | 134 | 1.93 × |
| FedRep | - | - | - | - | 116 | 2.25 × | 117 | 2.21 × |
| FedBABU | 156 | 1.14 × | 174 | 1.22 × | 178 | 1.47 × | 181 | 1.43 × |
| Fed-RoD | **72** | **2.47 ×** | 92 | 2.30 × | 132 | 1.98 × | 95 | 2.72 × |
| Ditto | 178 | 1.00 × | 212 | 1.00 × | 261 | 1.00 × | - | - |
| DFedAvgM | 115 | 1.55 × | 136 | 1.56 × | 160 | 1.63 × | 258 | 1.00 × |
| Dis-PFL | 143 | 1.24 × | 166 | 1.28 × | 227 | 1.15 × | 188 | 1.37 × |
| DFedSAM | 158 | 1.13 × | 174 | 1.22 × | 229 | 1.14 × | 214 | 1.21 × |
| DFedMDC | 74 | 2.41 × | 108 | 1.96 × | **54** | **4.83 ×** | **53** | **4.87 ×** |
| DFedSMDC | 82 | 2.17 × | **78** | **2.72 ×** | 70 | 3.73 × | 73 | 3.53 × |

## C  PROOF OF THEORETICAL ANALYSIS

### C.1  PRELIMINARY LEMMAS

**Lemma 1** (Lemma 4, (Lian et al., 2017))**.** *For any $t \in \mathbb{Z}^{+}$, the mixing matrix $\mathbf{W} \in \mathbb{R}^{m}$ satisfies $\|\mathbf{W}^t - \mathbf{P}\|_{\mathrm{op}} \leq \lambda^t$, where $\lambda := \max\{|\lambda_2|, |\lambda_m(W)|\}$ and for a matrix $\mathbf{A}$, we denote its spectral norm as $\|\mathbf{A}\|_{\mathrm{op}}$. Furthermore, $\mathbf{1} := [1, 1, \ldots, 1]^{\top} \in \mathbb{R}^{m}$ and*

$$\mathbf{P} := \frac{\mathbf{1}\mathbf{1}^{\top}}{m} \in \mathbb{R}^{m \times m}.$$

**Lemma 2** (Lemma 23, (Pillutla et al., 2022))**.** *Consider $F$ which is $L$-smooth and fix a $v^0 \in \mathbb{R}^d$. Define the sequence $(v^k)$ of iterates produced by stochastic gradient descent with a fixed learning*

rate $\eta_v \leq 1/(2K_v L_v)$ *starting from* $v^0$, *we have the bound*

$$\mathbb{E}\|v^{K_v-1} - v^0\|^2 \leq 16\eta_v^2 K_v^2 \mathbb{E}\|\nabla F(v^0)\|^2 + 8\eta_v^2 K_v^2 \sigma_v^2.$$

**Lemma 3** (Local update for shared model $u_i$ in DFedMDC). *Assume that assumptions 1-3 hold, for all clients* $i \in \{1, 2, ..., m\}$ *and local iteration steps* $k \in \{0, 1, ..., K_u - 1\}$, *we can get*

$$\frac{1}{m} \sum_{i=1}^{m} \mathbb{E}\|u_i^{t,k} - u_i^t\|^2 \leq 18\eta_u^2 K_u^2 \Big(\sigma_u^2 + \delta^2 + \frac{1}{m} \sum_{i=1}^{m} \mathbb{E}\|\nabla_u F(u_i^t, V^{t+1})\|^2\Big).$$

*Proof.*

$$\frac{1}{m} \sum_{i=1}^{m} \mathbb{E}\|u_i^{t,k+1} - u_i^t\|^2 \leq \frac{1}{m} \sum_{i=1}^{m} \mathbb{E}\|u_i^{t,k} - \eta_u \nabla_u F_i(u_i^{t,k}, v_i^{t+1}; \xi_i) - u_i^t\|^2$$

$$\leq \frac{1}{m} \sum_{i=1}^{m} \mathbb{E}\Big\|u_i^{t,k} - u_i^t - \eta_u \Big(\nabla_u F_i(u_i^{t,k}, v_i^{t+1}; \xi_i) - \nabla_u F_i(u_i^{t,k}, v_i^{t+1}) + \nabla_u F_i(u_i^{t,k}, v_i^{t+1})$$

$$- \nabla_u F_i(u_i^t, V^{t+1}) + \nabla_u F_i(u_i^t, V^{t+1})\Big)\Big\|^2$$

$$\leq \text{I} + \text{II}.$$

Where

$$\text{I} = (1 + \frac{1}{2K_u - 1}) \frac{1}{m} \sum_{i=1}^{m} \mathbb{E}\|u_i^{t,k} - u_i^t\|^2,$$

and

$$\text{II} = \frac{2K_u^2 \eta_u^2}{m} \sum_{i=1}^{m} \mathbb{E}\Big\|\nabla_u F_i(u_i^{t,k}, v_i^{t+1}; \xi_i) - \nabla_u F_i(u_i^{t,k}, v_i^{t+1}) + \nabla_u F_i(u_i^{t,k}, v_i^{t+1}) - \nabla_u F(u_i^t, V^{t+1}) + \nabla_u F(u_i^t, V^{t+1})\Big)\Big\|^2.$$

For II, we use assumptions 2-3 and generate the following:

$$\text{II} = 6\eta_u^2 K_u \Big(\sigma_u^2 + \delta^2 + \frac{1}{m} \sum_{i=1}^{m} \mathbb{E}\big\|\nabla_u F(u_i^t, V^{t+1})\big\|^2\Big).$$

Therefore, the recursion from $j = 0$ to $K_u - 1$ can generate:

$$\frac{1}{m} \sum_{i=1}^{m} \mathbb{E}\|u_i^{t,k} - u_i^t\|^2 \leq \sum_{j=0}^{K_u-1} (1 + \frac{1}{2K_u - 1})^j \text{II}$$

$$\leq (2K_u - 1)\Big[(1 + \frac{1}{2K_u - 1})_u^K - 1\Big]\text{II}$$

$$\overset{a)}{\leq} 3K_u \text{II}$$

$$\leq 18\eta_u^2 K_u^2 \Big(\sigma_u^2 + \delta^2 + \frac{1}{m} \sum_{i=1}^{m} \mathbb{E}\|\nabla_u F(u_i^t, V^{t+1})\|^2\Big),$$

where a) uses $1 + \frac{1}{2K_u - 1} \leq 2$ and $(1 + \frac{1}{2K_u - 1})^{2K_u \cdot \frac{1}{2}} \leq \sqrt{5} < \frac{5}{2}$ for any $K_u \geq 1$. $\square$

**Lemma 4** (Local update for shared model $u_i$ in DFedSMDC). *Assume that assumptions 1-3 hold, for all clients* $i \in \{1, 2, ..., m\}$ *and local iteration steps* $k \in \{0, 1, ..., K_u - 1\}$, *we can get*

$$\frac{1}{m} \sum_{i=1}^{m} \mathbb{E}\|u_i^{t,k} - u_i^t\|^2 \leq 6\eta_u^2 K_u L_u^2 \rho^2 + 18\eta_u^2 K_u^2 \Big(\sigma_u^2 + \delta^2 + \frac{1}{m} \sum_{i=1}^{m} \mathbb{E}\|\nabla_u F(u_i^t, V^{t+1})\|^2\Big).$$

*Proof.*

$$\frac{1}{m}\sum_{i=1}^{m}\mathbb{E}\big\|u_i^{t,k+1}-u_i^t\big\|^2 \leq \frac{1}{m}\sum_{i=1}^{m}\mathbb{E}\big\|u_i^{t,k}-\eta_u\nabla_u F_i(u_i^{t,k}+\epsilon(u_i^{t,k}),v_i^{t+1};\xi_i)-u_i^t\big\|^2$$

$$\leq \frac{1}{m}\sum_{i=1}^{m}\mathbb{E}\big\|u_i^{t,k}-u_i^t-\eta_u\Big(\nabla_u F_i(u_i^{t,k}+\epsilon(u_i^{t,k}),v_i^{t+1};\xi_i)-\nabla_u F_i(u_i^{t,k},v_i^{t+1};\xi_i)$$

$$+\nabla_u F_i(u_i^{t,k},v_i^{t+1};\xi_i)-\nabla_u F_i(u_i^{t,k},v_i^{t+1})+\nabla_u F_i(u_i^{t,k},v_i^{t+1})$$

$$-\nabla_u F_i(u_i^t,V^{t+1})+\nabla_u F_i(u_i^t,V^{t+1})\Big)\big\|^2$$

$$\leq \text{I}+\text{II}.$$

Where

$$\text{I}=(1+\frac{1}{2K_u-1})\frac{1}{m}\sum_{i=1}^{m}\mathbb{E}\big\|u_i^{t,k}-u_i^t-\eta_u\Big(\nabla_u F_i(u_i^{t,k}+\epsilon(u_i^{t,k}),v_i^{t+1};\xi_i)-\nabla_u F_i(u_i^{t,k},v_i^{t+1};\xi_i)\Big)\big\|^2,$$

and

$$\text{II}=\frac{2K_u^2\eta_u^2}{m}\sum_{i=1}^{m}\mathbb{E}\big\|\nabla_u F_i(u_i^{t,k},v_i^{t+1};\xi_i)-\nabla_u F_i(u_i^{t,k},v_i^{t+1})+\nabla_u F_i(u_i^{t,k},v_i^{t+1})-\nabla_u F(u_i^t,V^{t+1})+\nabla_u F(u_i^t,V^{t+1})\Big)\big\|^2.$$

For I, we use assumption 1 and generate the following:

$$\text{I}\leq (1+\frac{1}{2K_u-1})\frac{1}{m}\sum_{i=1}^{m}\Big(\mathbb{E}\big\|u_i^{t,k}-u_i^t\big\|^2+\eta_u^2 L_u^2\mathbb{E}\big\|\rho\frac{\nabla_u F_i(u_i^{t,k},v_i^{t+1};\xi_i)}{\|\nabla_u F_i(u_i^{t,k},v_i^{t+1};\xi_i)\|_2}\big\|^2\Big)$$

$$\leq (1+\frac{1}{2K_u-1})\Big(\frac{1}{m}\sum_{i=1}^{m}\mathbb{E}\big\|u_i^{t,k}-u_i^t\big\|^2+\eta_u^2 L_u^2\rho^2\Big).$$

For II, we use assumption 2-3 and generate the following:

$$\text{II}=6\eta_u^2 K_u\Big(\sigma_u^2+\delta^2+\frac{1}{m}\sum_{i=1}^{m}\mathbb{E}\big\|\nabla_u F(u_i^t,V^{t+1})\big\|^2\Big).$$

Therefore, the recursion from $j=0$ to $K_u-1$ can generate:

$$\frac{1}{m}\sum_{i=1}^{m}\mathbb{E}\big\|u_i^{t,k}-u_i^t\big\|^2 \leq \sum_{j=0}^{K_u-1}(1+\frac{1}{2K_u-1})^j\Big[(1+\frac{1}{2K_u-1})\eta_u^2 L_u^2\rho^2+\text{II}\Big]$$

$$\leq (2K_u-1)\Big[(1+\frac{1}{2K_u-1})^K_u-1\Big]\Big((1+\frac{1}{2K_u-1})\eta_u^2 L_u^2\rho^2+\text{II}\Big)$$

$$\overset{a)}{\leq} 3K_u(\text{II}+2\eta^2 L_u^2\rho^2)$$

$$\leq 6\eta_u^2 K_u L_u^2\rho^2+18\eta_u^2 K_u^2\Big(\sigma_u^2+\delta^2+\frac{1}{m}\sum_{i=1}^{m}\mathbb{E}\big\|\nabla_u F(u_i^t,V^{t+1})\big\|^2\Big),$$

where a) uses $1+\frac{1}{2K_u-1}\leq 2$ and $(1+\frac{1}{2K_u-1})^{2K_u\cdot\frac{1}{2}}\leq\sqrt{5}<\frac{5}{2}$ for any $K_u\geq 1$. $\qquad\square$

**Lemma 5** (Shared model shift in DFedMDC)**.** *Assume that assumptions 1-3 hold, for all clients $i\in\{1,2,...,m\}$ and local iteration steps $k\in\{1,2,...,K_u\}$, we can get*

$$\frac{1}{m}\sum_{i=1}^{m}\mathbb{E}\|u_i^t-\bar{u}^t\|^2\leq\frac{18\eta_u^2 K_u^2}{(1-\lambda)^2}\Big(\sigma_u^2+\delta^2+\frac{1}{m}\sum_{i=1}^{m}\mathbb{E}\|\nabla_u F(u_i^t,V^{t+1})\|^2\Big).$$

*Proof.* Inspired by Lemma D.2 in (Shi et al., 2023b) and Lemma 4 in (Sun et al., 2022), according to Lemma 1, we can generate

$$
\begin{aligned}
\mathbb{E}\|U^t - \mathbf{P}U^t\|^2 &\leq \mathbb{E}\|\sum_{j=0}^{t-1}(\mathbf{P} - \mathbf{W}^{t-1-j})\zeta^j\|^2 \\
&\leq \sum_{j=0}^{t-1}\|\mathbf{P} - \mathbf{W}^{t-1-j}\|_{\mathrm{op}}\|\zeta^j\| \\
&\leq (\sum_{j=0}^{t-1}\lambda^{t-1-j}\|\zeta^j\|)^2 \\
&\leq \mathbb{E}(\sum_{j=0}^{t-1}\lambda^{\frac{t-1-j}{2}}\cdot\lambda^{\frac{t-1-j}{2}}\|\zeta^j\|)^2 \\
&\leq (\sum_{j=0}^{t-1}\lambda^{t-1-j})(\sum_{j=0}^{t-1}\lambda^{t-1-j}\mathbb{E}\|\zeta^j\|^2),
\end{aligned}
$$

where $U^t = [u_1^t, u_2^t, ..., u_m^t]^T \in \mathbb{R}^{m \times d}$ and

$$
\mathbb{E}\|\zeta^j\|^2 \leq \|\mathbf{W}\|^2 \cdot \mathbb{E}\|U^j - Z^j\|^2 \leq \mathbb{E}\|U^j - Z^j\|^2.
$$

Note that $Z^t = [z_1^t, z_2^t, ..., z_m^t]^T \in \mathbb{R}^{m \times d}$. According to Lemma 3, for any $j$, we have

$$
\mathbb{E}\|U^j - Z^j\|^2 \leq 18\eta_u^2 K_u^2 m\Big(\sigma_u^2 + \delta^2 + \frac{1}{m}\sum_{i=1}^m \mathbb{E}\big\|\nabla_u F(u_i^t, V^{t+1})\big\|^2\Big).
$$

After that,

$$
\mathbb{E}\|U^t - \mathbf{P}U^t\|^2 \leq \frac{18\eta_u^2 K_u^2}{(1-\lambda)^2}\Big(\sigma_u^2 + \delta^2 + \frac{1}{m}\sum_{i=1}^m \mathbb{E}\|\nabla_u F(u_i^t, V^{t+1})\|^2\Big).
$$

The fact is that $U^t - \mathbf{P}U^t = \begin{pmatrix} u_1^t - \bar{u}^t \\ u_2^t - \bar{u}^t \\ \vdots \\ u_m^t - \bar{u}^t \end{pmatrix}$ is the result needed to prove. $\qquad\square$

**Lemma 6** (Shared model shift in DFedSMDC). *Assume that assumptions 1-3 hold, for all clients $i \in \{1, 2, ..., m\}$ and local iteration steps $k \in \{1, 2, ..., K_u\}$, we can get*

$$
\frac{1}{m}\sum_{i=1}^m \mathbb{E}\|u_i^t - \bar{u}^t\|^2 \leq \frac{6\eta_u^2 K_u}{(1-\lambda)^2}\Big[L_u^2 \rho^2 + 3K_u\Big(\sigma_u^2 + \delta^2 + \frac{1}{m}\sum_{i=1}^m \mathbb{E}\|\nabla_u F(u_i^t, V^{t+1})\|^2\Big)\Big].
$$

*Proof.* Based on Lemma 5 and according to Lemma 4, for any $j$, we have

$$
\mathbb{E}\|U^j - Z^j\|^2 \leq m\Big(6\eta_u^2 K_u L_u^2 \rho^2 + 18\eta_u^2 K_u^2\Big(\sigma_u^2 + \delta^2 + \frac{1}{m}\sum_{i=1}^m \mathbb{E}\big\|\nabla_u F(u_i^t, V^{t+1})\big\|^2\Big)\Big).
$$

After that,

$$
\mathbb{E}\|U^t - \mathbf{P}U^t\|^2 \leq \frac{6\eta_u^2 K_u}{(1-\lambda)^2}\Big[L_u^2 \rho^2 + 3K_u\Big(\sigma_u^2 + \delta^2 + \frac{1}{m}\sum_{i=1}^m \mathbb{E}\|\nabla_u F(u_i^t, V^{t+1})\|^2\Big)\Big].
$$

The fact is that $U^t - \mathbf{P}U^t = \begin{pmatrix} u_1^t - \bar{u}^t \\ u_2^t - \bar{u}^t \\ \vdots \\ u_m^t - \bar{u}^t \end{pmatrix}$ is the result needed to prove. $\qquad\square$

## C.2 Proof of Convergence Analysis

**Proof Outline and the Challenge of Dependent Random Variables.** We start with

$$
\begin{aligned}
F\left(\bar{u}^{t+1}, V^{t+1}\right) - F\left(\bar{u}^t, V^t\right) = {} & F\left(\bar{u}^t, V^{t+1}\right) - F\left(\bar{u}^t, V^t\right) \\
& + F\left(\bar{u}^{t+1}, V^{t+1}\right) - F\left(\bar{u}^t, V^{t+1}\right).
\end{aligned}
\tag{5}
$$

The first line corresponds to the effect of the $v$-step and the second line to the $u$-step. The former is

$$
\begin{aligned}
F\left(\bar{u}^t, V^{t+1}\right) - F\left(\bar{u}^t, V^t\right) = {} & \frac{1}{m} \sum_{i=1}^m \mathbb{E}\Big[ F_i(\bar{u}^t, v_i^{t+1}) - F_i(\bar{u}^t, v_i^t) \Big] \\
\leq {} & \frac{1}{m} \sum_{i=1}^m \mathbb{E}\Big[ \big\langle \nabla_v F_i\left(\bar{u}^t, v_i^t\right), v_i^{t+1} - v_i^t \big\rangle + \frac{L_v}{2} \|v_i^{t+1} - v_i^t\|^2 \Big].
\end{aligned}
\tag{6}
$$

It is easy to handle with standard techniques that rely on the smoothness of $F\left(u^t, \cdot\right)$. The latter is more challenging. In particular, the smoothness bound for the $u$-step gives us

$$
F\left(\bar{u}^{t+1}, V^{t+1}\right) - F\left(\bar{u}^t, V^{t+1}\right) \leq \big\langle \nabla_u F\left(\bar{u}^t, V^{t+1}\right), \bar{u}^{t+1} - \bar{u}^t \big\rangle + \frac{L_u}{2} \|\bar{u}^{t+1} - \bar{u}^t\|^2.
\tag{7}
$$

### C.2.1 Proof of Convergence Analysis for DFedMDC

**Analysis of the $u$-Step.**

$$
\begin{aligned}
\mathbb{E}\Big[ F\left(\bar{u}^{t+1}, V^{t+1}\right) - F\left(\bar{u}^t, V^{t+1}\right) \Big] & \leq \big\langle \nabla_u F\left(\bar{u}^t, V^{t+1}\right), \bar{u}^{t+1} - \bar{u}^t \big\rangle + \frac{L_u}{2} \mathbb{E}\|\bar{u}^{t+1} - \bar{u}^t\|^2 \\
& \leq \frac{-\eta_u}{m} \sum_{i=1}^m \mathbb{E}\Big\langle \nabla_u F\left(\bar{u}^t, V^{t+1}\right), \sum_{k=0}^{K_u-1} \nabla_u F\left(u_i^{t,k}, v_i^{t+1}; \xi_i\right) \Big\rangle + \frac{L_u}{2} \mathbb{E}\|\bar{u}^{t+1} - \bar{u}^t\|^2 \\
& \leq -\eta_u K_u \mathbb{E}[\Delta_{\bar{u}}^t] + \frac{\eta_u}{m} \sum_{i=1}^m \sum_{k=0}^{K_u-1} \mathbb{E}\Big\langle \nabla_u F\left(\bar{u}^t, V^{t+1}\right), \nabla F\left(\bar{u}^t, v_i^{t+1}\right) - \nabla_u F\left(u_i^{t,k}, v_i^{t+1}; \xi_i\right) \Big\rangle + \frac{L_u}{2} \mathbb{E}\|\bar{u}^{t+1} - \bar{u}^t\|^2 \\
& \overset{a)}{\leq} \frac{-\eta_u K_u}{2} \mathbb{E}[\Delta_{\bar{u}}^t] + \underbrace{\frac{\eta_u L_u^2}{2m} \sum_{i=1}^m \sum_{k=0}^{K_u-1} \mathbb{E}\|u_i^{t,k} - \bar{u}^t\|^2}_{\mathcal{T}_{1,u}} + \underbrace{\frac{L_u}{2} \mathbb{E}\|\bar{u}^{t+1} - \bar{u}^t\|^2}_{\mathcal{T}_{2,u}}.
\end{aligned}
\tag{8}
$$

Where a) uses $\mathbb{E}\left[\nabla_u F(u_i^{t,k}, v_i^{t+1}; \xi_i)\right] = \nabla_u F\left(u_i^{t,k}, v_i^{t+1}\right)$ and $\langle x, y \rangle \leq \frac{1}{2}\|x\|^2 + \frac{1}{2}\|y\|^2$ for vectors $x, y$ followed by $L_u$-smoothness.
For $\mathcal{T}_{1,u}$, we can use Lemma 5.

$$
\mathcal{T}_{1,u} \leq \frac{9\eta_u^3 K_u^2 L_u^2}{(1-\lambda)^2} \Bigg[ \sigma_u^2 + \delta^2 + \underbrace{\frac{1}{m} \sum_{i=1}^m \mathbb{E}\big\| \nabla_u F(u_i^t, V^{t+1}) \big\|^2}_{\mathcal{T}_{3,u}} \Bigg]
\tag{9}
$$

For $\mathcal{T}_{3,u}$,

$$
\begin{aligned}
\mathcal{T}_{3,u} & \leq \frac{1}{m} \sum_{i=1}^m \mathbb{E}\big\| \nabla_u F(u_i^t, V^{t+1}) - \nabla_u F(\bar{u}^t, V^{t+1}) + \nabla_u F(\bar{u}^t, V^{t+1}) \big\|^2 \\
& \leq \frac{2L_u^2}{m} \sum_{i=1}^m \mathbb{E}\|u_i^t - \bar{u}^t\|^2 + \frac{2}{m} \sum_{i=1}^m \mathbb{E}\|\nabla_u F(\bar{u}^t, V^{t+1})\|^2 \\
& \leq \frac{2L_u^2}{m} \sum_{i=1}^m \mathbb{E}\|u_i^t - \bar{u}^t\|^2 + 2\mathbb{E}[\Delta_{\bar{u}}^t],
\end{aligned}
\tag{10}
$$

After that, combining Eq. (9) and (10) and assuming local learning rate $\eta_u \ll \frac{1-\lambda}{3\sqrt{2K_u L_u}}$, we can generate

$$\mathcal{T}_{1,u} \le \frac{9\eta_u^3 K_u^2 L_u^2}{(1-\lambda)^2} \left[ \sigma_u^2 + \delta^2 + 2\mathbb{E}[\Delta_{\bar{u}}^t] \right]. \tag{11}$$

Meanwhile, for $\mathcal{T}_{2,u}$,

$$
\begin{aligned}
\mathcal{T}_{2,u} &\le \frac{\eta_u^2 L_u}{2m} \sum_{i=1}^m \sum_{k=0}^{K_u-1} \Big\| \nabla_u F\left(u_i^{t,k}, v_i^{t+1}; \xi_i\right) - \nabla_u F\left(u_i^t, v_i^{t+1}\right) + \nabla_u F\left(u_i^t, v_i^{t+1}\right) \\
&\quad - \nabla_u F\left(u_i^t, V^{t+1}\right) + \nabla_u F\left(u_i^t, V^{t+1}\right) - \nabla_u F\left(\bar{u}^t, V^{t+1}\right) + \nabla_u F\left(\bar{u}^t, V^{t+1}\right) \Big\|^2 \\
&\le 2\eta_u^2 K_u L_u \Big( \sigma_u^2 + \delta^2 + \frac{L_u^2}{m} \sum_{i=1}^m \mathbb{E}\|u_i^t - \bar{u}^t\|^2 + \mathbb{E}[\Delta_{\bar{u}}^t] \Big) \\
&\le 2\eta_u^2 K_u L_u \Big( \sigma_u^2 + \delta^2 + \mathbb{E}[\Delta_{\bar{u}}^t] \Big) + \underbrace{\frac{2\eta_u^2 K_u L_u^3}{m} \sum_{i=1}^m \mathbb{E}\|u_i^t - \bar{u}^t\|^2}_{\mathcal{T}_{4,u}}
\end{aligned} \tag{12}
$$

For $\mathcal{T}_{4,u}$, we can use Lemma 5.

After that,

$$
\begin{aligned}
\mathbb{E}\Big[ F\left(\bar{u}^{t+1}, V^{t+1}\right) - F\left(\bar{u}^t, V^{t+1}\right) \Big] &\le \frac{-\eta_u K_u}{2} \mathbb{E}[\Delta_{\bar{u}}^t] + \mathcal{T}_{1,u} + \mathcal{T}_{2,u} \\
&\le \Big( \frac{-\eta_u K_u}{2} + 2\eta_u^2 K_u L_u + \frac{18\eta_u^2 K_u^2(2+\eta_u L_u^2)}{(1-\lambda)^2} \Big) \mathbb{E}[\Delta_{\bar{u}}^t] \\
&\quad + 2\eta_u^2 K_u L_u(\sigma_u^2 + \delta^2) + \frac{9\eta_u^2 K_u^2(2+\eta_u L_u^2)}{(1-\lambda)^2} \Big( \sigma_u^2 + \delta^2 \Big).
\end{aligned} \tag{13}
$$

**Analysis of the $v$-Step.**

$$\mathbb{E}\Big[ F\left(\bar{u}^t, V^{t+1}\right) - F\left(\bar{u}^t, V^t\right) \Big] \le \underbrace{\frac{1}{m} \sum_{i=1}^m \mathbb{E}\Big\langle \nabla_v F_i\left(\bar{u}^t, v_i^t\right), v_i^{t+1} - v_i^t \Big\rangle}_{\mathcal{T}_{1,v}} + \underbrace{\frac{L_v}{2m} \sum_{i=1}^m \mathbb{E}\|v_i^{t+1} - v_i^t\|^2}_{\mathcal{T}_{2,v}}. \tag{14}$$

For $\mathcal{T}_{1,v}$,

$$
\begin{aligned}
\mathcal{T}_{1,v} &\le \frac{1}{m} \sum_{i=1}^m \mathbb{E}\Big\langle \nabla_v F_i\left(\bar{u}^t, v_i^t\right) - \nabla_v F_i\left(u_i^t, v_i^t\right) + \nabla_v F_i\left(u_i^t, v_i^t\right), -\eta_v \sum_{k=0}^{K_v-1} \mathbb{E}\nabla_v F_i(u_i^t, v_i^t; \xi_i) \Big\rangle \\
&\overset{a)}{\le} \frac{-\eta_v K_v}{m} \sum_{i=1}^m \mathbb{E}\|\nabla_v F_i(u_i^t, v_i^t)\|^2 + \frac{1}{m} \sum_{i=1}^m \mathbb{E}\Big\langle \nabla_v F_i\left(\bar{u}^t, v_i^t\right) - \nabla_v F_i\left(u_i^t, v_i^t\right), v_i^{t+1} - v_i^t \Big\rangle \\
&\overset{b)}{\le} -\eta_v K_v \mathbb{E}[\Delta_v^t] + \underbrace{\frac{L_{vu}^2}{2m} \sum_{i=1}^m \mathbb{E}\|\bar{u}^t - u_i^t\|^2}_{\mathcal{T}_{3,v}} + \underbrace{\frac{1}{2m} \sum_{i=1}^m \mathbb{E}\|v_i^{t+1} - v_i^t\|^2}_{\frac{1}{L_v}\mathcal{T}_{2,v}},
\end{aligned} \tag{15}
$$

where a) and b) is get from the unbiased expectation property of $\nabla_v F_i(u_i^t, v_i^t; \xi_i)$ and $<x,y> \le \frac{1}{2}(\|x\|^2 + \|y\|^2)$, respectively.

For $\mathcal{T}_{2,v}$, according to Lemma 2, we have

$$
\begin{aligned}
\mathcal{T}_{2,v} &\le \frac{L_v}{2} \Big( \frac{16\eta_v^2 K_v^2}{m} \sum_{i=1}^m \mathbb{E}\|\nabla_v F_i(u_i^t, v_i^t)\|^2 + 8\eta_v^2 K_v^2 \sigma_v^2 \Big) \\
&\le 8L_v \eta_v^2 K_v^2 \mathbb{E}[\Delta_v^t] + 4L_v \eta_v^2 K_v^2 \sigma_v^2.
\end{aligned} \tag{16}
$$

For $\mathcal{T}_{3,v}$, according to Eq. (11), we have

$$\frac{L_{vu}^2}{2m} \sum_{i=1}^m \mathbb{E}\|\bar{u}^t - u_i^t\|^2 \leq \frac{L_{vu}^2}{(1-\lambda)^2}\left[18\eta_u^2 K_u^2\Big(\sigma_u^2 + \delta^2 + 2\mathbb{E}[\Delta_{\bar{u}}^t]\Big)\right]. \tag{17}$$

After that, summing Eq. (15), (16), and (17), we have

$$\mathbb{E}\Big[F\left(\bar{u}^t, V^{t+1}\right) - F\left(\bar{u}^t, V^t\right)\Big] \leq \Big(-\eta_v K_v + 8\eta_v^2 K_v^2(L_v + 1)\Big)\mathbb{E}[\Delta_v^t] + 4\eta_v^2 K_v^2 \sigma_v^2(L_v + 1)$$
$$+ \frac{L_{vu}^2}{(1-\lambda)^2}\left[18\eta_u^2 K_u^2\Big(\sigma_u^2 + \delta^2 + 2\mathbb{E}[\Delta_{\bar{u}}^t]\Big)\right]. \tag{18}$$

**Obtaining the Final Convergence Bound.**

$$\mathbb{E}\Big[F\left(\bar{u}^{t+1}, V^{t+1}\right) - F\left(\bar{u}^t, V^t\right)\Big] = \mathbb{E}\Big[F\left(\bar{u}^t, V^{t+1}\right) - F\left(\bar{u}^t, V^t\right) + F\left(\bar{u}^{t+1}, V^{t+1}\right) - F\left(\bar{u}^t, V^{t+1}\right)\Big]$$
$$\leq \Big(\frac{-\eta_u K_u}{2} + 2\eta_u^2 K_u L_u + \frac{18\eta_u^2 K_u^2(2 + \eta_u L_u^2)}{(1-\lambda)^2}\Big)\mathbb{E}[\Delta_{\bar{u}}^t]$$
$$+ 2\eta_u^2 K_u L_u(\sigma_u^2 + \delta^2) + \frac{9\eta_u^2 K_u^2(2 + \eta_u L_u^2)}{(1-\lambda)^2}\Big(\sigma_u^2 + \delta^2\Big)$$
$$+ \Big(-\eta_v K_v + 8\eta_v^2 K_v^2(L_v + 1)\Big)\mathbb{E}[\Delta_v^t] + 4\eta_v^2 K_v^2 \sigma_v^2(L_v + 1)$$
$$+ \frac{18\eta_u^2 L_{vu}^2 K_u^2\Big(\sigma_u^2 + \delta^2 + 2\mathbb{E}[\Delta_{\bar{u}}^t]\Big)}{(1-\lambda)^2}. \tag{19}$$

Summing from $t = 1$ to $T$, assume the local learning rates satisfy $\eta_u = \mathcal{O}(1/L_u K_u \sqrt{T}), \eta_v = \mathcal{O}(1/L_v K_v \sqrt{T})$, $F^*$ is denoted as the minimal value of $F$, i.e., $F(\bar{u}, V) \geq F^*$ for all $\bar{u} \in \mathbb{R}^d$, and $V = (v_1, \ldots, v_m) \in \mathbb{R}^{d_1 + \ldots + d_m}$. We can generate

$$\frac{1}{T}\sum_{i=1}^T\Big(\frac{1}{L_u}\mathbb{E}[\Delta_{\bar{u}}^t] + \frac{1}{L_v}\mathbb{E}[\Delta_v^t]\Big) \leq \mathcal{O}\Big(\frac{F(\bar{u}^1, V^1) - F^*}{\sqrt{T}} + \frac{\sigma_v^2(L_v + 1)}{L_v^2\sqrt{T}} + \frac{(L_{vu}^2 + 1)(\sigma_u^2 + \delta^2)}{L_u^2(1-\lambda)^2\sqrt{T}}$$
$$+ \frac{\sigma_u^2 + \delta^2}{K_u L_u \sqrt{T}} + \frac{\sigma_u^2 + \delta^2}{K_u L_u(1-\lambda)^2 T}\Big). \tag{20}$$

Assume that

$$\sigma_1^2 = \frac{\chi^2 L_v(\sigma_u^2 + \delta^2)}{L_u} + \frac{\sigma_u^2 + \delta^2}{L_u^2}, \quad \sigma_2^2 = \frac{\sigma_v^2(L_v + 1)}{L_v^2} + \frac{\sigma_u^2 + \delta^2}{K_u L_u}, \quad \sigma_3^2 = \frac{\sigma_u^2 + \delta^2}{K_u L_u}.$$

Then, we have the final convergence bound:

$$\frac{1}{T}\sum_{i=1}^T\Big(\frac{1}{L_u}\mathbb{E}[\Delta_{\bar{u}}^t] + \frac{1}{L_v}\mathbb{E}[\Delta_v^t]\Big) \leq \mathcal{O}\Big(\frac{F(\bar{u}^1, V^1) - F^*}{\sqrt{T}} + \frac{\sigma_1^2}{(1-\lambda)^2\sqrt{T}} + \frac{\sigma_2^2}{\sqrt{T}} + \frac{\sigma_3^2}{(1-\lambda)^2 T}\Big). \tag{21}$$

### C.2.2 Proof of Convergence Analysis for DFedSMDC

**Analysis of the $u$-Step.**

$$\mathbb{E}\Big[F\left(\bar{u}^{t+1}, V^{t+1}\right) - F\left(\bar{u}^t, V^{t+1}\right)\Big] \leq \Big\langle \nabla_u F\left(\bar{u}^t, V^{t+1}\right), \bar{u}^{t+1} - \bar{u}^t \Big\rangle + \frac{L_u}{2}\mathbb{E}\|\bar{u}^{t+1} - \bar{u}^t\|^2$$

$$\leq \frac{-\eta_u}{m}\sum_{i=1}^{m}\mathbb{E}\Big\langle \nabla_u F\left(\bar{u}^t, V^{t+1}\right), \sum_{k=0}^{K_u-1}\nabla_u F\left(u_i^{t,k}, v_i^{t+1}; \xi_i\right)\Big\rangle + \frac{L_u}{2}\mathbb{E}\|\bar{u}^{t+1} - \bar{u}^t\|^2$$

$$\leq -\eta_u K_u \mathbb{E}[\Delta_{\bar{u}}^t] + \frac{\eta_u}{m}\sum_{i=1}^{m}\sum_{k=0}^{K_u-1}\mathbb{E}\Big\langle \nabla_u F\left(\bar{u}^t, V^{t+1}\right), \nabla F\left(\bar{u}^t, v_i^{t+1}\right) - \nabla_u F\left(u_i^{t,k}, v_i^{t+1}; \xi_i\right)\Big\rangle + \frac{L_u}{2}\mathbb{E}\|\bar{u}^{t+1} - \bar{u}^t\|^2$$

$$\overset{a)}{\leq} \frac{-\eta_u K_u}{2}\mathbb{E}[\Delta_{\bar{u}}^t] + \underbrace{\frac{\eta_u L_u^2}{2m}\sum_{i=1}^{m}\sum_{k=0}^{K_u-1}\mathbb{E}\|u_i^{t,k} - \bar{u}^t\|^2}_{\mathcal{T}_{1,u}} + \underbrace{\frac{L_u}{2}\mathbb{E}\|\bar{u}^{t+1} - \bar{u}^t\|^2}_{\mathcal{T}_{2,u}}.$$

$$\tag{22}$$

Where a) uses $\mathbb{E}\Big[\nabla_u F(u_i^{t,k}, v_i^{t+1}; \xi_i)\Big] = \nabla_u F\left(u_i^{t,k}, v_i^{t+1}\right)$ and $\langle x, y\rangle \leq \frac{1}{2}\|x\|^2 + \frac{1}{2}\|y\|^2$ for vectors $x, y$ followed by $L_u$-smoothness.

For $\mathcal{T}_{1,u}$, we can use Lemma 6.

$$\mathcal{T}_{1,u} \leq \frac{3\eta_u^3 K_u L_u^2}{(1-\lambda)^2}\left[L_u^2\rho^2 + 3K_u\Big(\sigma_u^2 + \delta^2 + \underbrace{\frac{1}{m}\sum_{i=1}^{m}\mathbb{E}\big\|\nabla_u F(u_i^t, V^{t+1})\big\|^2}_{\mathcal{T}_{3,u}}\Big)\right] \tag{23}$$

For $\mathcal{T}_{3,u}$,

$$\mathcal{T}_{3,u} \leq \frac{1}{m}\sum_{i=1}^{m}\mathbb{E}\big\|\nabla_u F(u_i^t, V^{t+1}) - \nabla_u F(\bar{u}^t, V^{t+1}) + \nabla_u F(\bar{u}^t, V^{t+1})\big\|^2$$

$$\leq \frac{2L_u^2}{m}\sum_{i=1}^{m}\mathbb{E}\|u_i^t - \bar{u}^t\|^2 + \frac{2}{m}\sum_{i=1}^{m}\mathbb{E}\|\nabla_u F(\bar{u}^t, V^{t+1})\|^2 \tag{24}$$

$$\leq \frac{2L_u^2}{m}\sum_{i=1}^{m}\mathbb{E}\|u_i^t - \bar{u}^t\|^2 + 2\mathbb{E}[\Delta_{\bar{u}}^t],$$

After that, combining Eq. (23) and (24) and assuming local learning rate $\eta_u \ll \frac{1-\lambda}{3\sqrt{2K_u L_u}}$, we can generate

$$\mathcal{T}_{1,u} \leq \frac{3\eta_u^3 K_u L_u^2}{(1-\lambda)^2}\left[L_u^2\rho^2 + 3K_u\Big(\sigma_u^2 + \delta^2 + 2\mathbb{E}[\Delta_{\bar{u}}^t]\Big)\right]. \tag{25}$$

Meanwhile, for $\mathcal{T}_{2,u}$,

$$\mathcal{T}_{2,u} \leq \frac{\eta_u^2 L_u}{2m}\sum_{i=1}^{m}\sum_{k=0}^{K_u-1}\Big\|\nabla_u F\left(u_i^{t,k} + \epsilon(u_i^{t,k}), v_i^{t+1}; \xi_i\right) - \nabla_u F\left(u_i^{t,k}, v_i^{t+1}; \xi_i\right)$$

$$+ \nabla_u F\left(u_i^{t,k}, v_i^{t+1}; \xi_i\right) - \nabla_u F\left(u_i^t, v_i^{t+1}\right) + \nabla_u F\left(u_i^t, v_i^{t+1}\right) - \nabla_u F\left(u_i^t, V^{t+1}\right) + \nabla_u F\left(u_i^t, V^{t+1}\right)$$

$$- \nabla_u F\left(\bar{u}^t, V^{t+1}\right) + \nabla_u F\left(\bar{u}^t, V^{t+1}\right)\Big\|^2$$

$$\leq \frac{5}{2}\eta_u^2 K_u L_u\Big(L_u^2\rho^2 + \sigma_u^2 + \delta^2 + \frac{L_u^2}{m}\sum_{i=1}^{m}\mathbb{E}\|u_i^t - \bar{u}^t\|^2 + \mathbb{E}[\Delta_{\bar{u}}^t]\Big)$$

$$\leq \frac{5}{2}\eta_u^2 K_u L_u\Big(L_u^2\rho^2 + \sigma_u^2 + \delta^2 + \mathbb{E}[\Delta_{\bar{u}}^t]\Big) + \underbrace{\frac{5\eta_u^2 K_u L_u^3}{2m}\sum_{i=1}^{m}\mathbb{E}\|u_i^t - \bar{u}^t\|^2}_{\mathcal{T}_{4,u}}$$

$$\tag{26}$$

For $\mathcal{T}_{4,u}$, we can use Lemma 6.

After that,

$$
\begin{aligned}
\mathbb{E}\Big[F\big(\bar{u}^{t+1}, V^{t+1}\big) - F\big(\bar{u}^t, V^{t+1}\big)\Big] &\leq \frac{-\eta_u K_u}{2}\mathbb{E}[\Delta_{\bar{u}}^t] + \mathcal{T}_{1,u} + \mathcal{T}_{2,u} \\
&\leq \Big(\frac{-\eta_u K_u}{2} + \frac{5}{2}\eta_u^2 K_u L_u + \frac{18\eta_u^3 K_u^2 L_u^2(1 + 5\eta_u K_u L_u)}{(1-\lambda)^2}\Big)\mathbb{E}[\Delta_{\bar{u}}^t] \\
&\quad + \frac{3\eta_u^3 K_u L_u^2}{(1-\lambda)^2}\Big[L_u^2\rho^2 + 3K_u\big(\sigma_u^2 + \delta^2\big)\Big] + \frac{5}{2}\eta_u^2 K_u L_u\big(L_u^2\rho^2 + \sigma_u^2 + \delta^2\big) \\
&\quad + \frac{15\eta_u^4 K_u^2 L_u^3}{(1-\lambda)^2}\Big[L_u^2\rho^2 + 3K_u\big(\sigma_u^2 + \delta^2\big)\Big].
\end{aligned}
\tag{27}
$$

**Analysis of the $v$-Step.**

$$
\mathbb{E}\Big[F\big(\bar{u}^t, V^{t+1}\big) - F\big(\bar{u}^t, V^t\big)\Big] \leq \underbrace{\frac{1}{m}\sum_{i=1}^m \mathbb{E}\Big\langle \nabla_v F_i\big(\bar{u}^t, v_i^t\big), v_i^{t+1} - v_i^t\Big\rangle}_{\mathcal{T}_{1,v}} + \underbrace{\frac{L_v}{2m}\sum_{i=1}^m \mathbb{E}\|v_i^{t+1} - v_i^t\|^2}_{\mathcal{T}_{2,v}}.
\tag{28}
$$

For $\mathcal{T}_{1,v}$,

$$
\begin{aligned}
\mathcal{T}_{1,v} &\leq \frac{1}{m}\sum_{i=1}^m \mathbb{E}\Big\langle \nabla_v F_i\big(\bar{u}^t, v_i^t\big) - \nabla_v F_i\big(u_i^t, v_i^t\big) + \nabla_v F_i\big(u_i^t, v_i^t\big), -\eta_v \sum_{k=0}^{K_v-1}\mathbb{E}\nabla_v F_i(u_i^t, v_i^t; \xi_i)\Big\rangle \\
&\overset{a)}{\leq} \frac{-\eta_v K_v}{m}\sum_{i=1}^m \mathbb{E}\|\nabla_v F_i(u_i^t, v_i^t)\|^2 + \frac{1}{m}\sum_{i=1}^m \mathbb{E}\Big\langle \nabla_v F_i\big(\bar{u}^t, v_i^t\big) - \nabla_v F_i\big(u_i^t, v_i^t\big), v_i^{t+1} - v_i^t\Big\rangle \\
&\overset{b)}{\leq} -\eta_v K_v \mathbb{E}[\Delta_v^t] + \underbrace{\frac{L_{vu}^2}{2m}\sum_{i=1}^m \mathbb{E}\|\bar{u}^t - u_i^t\|^2}_{\mathcal{T}_{3,v}} + \underbrace{\frac{1}{2m}\sum_{i=1}^m \mathbb{E}\|v_i^{t+1} - v_i^t\|^2}_{\frac{1}{L_v}\mathcal{T}_{2,v}},
\end{aligned}
\tag{29}
$$

where a) and b) is get from the unbiased expectation property of $\nabla_v F_i(u_i^t, v_i^t; \xi_i)$ and $<x, y> \leq \frac{1}{2}(\|x\|^2 + \|y\|^2)$, respectively.

For $\mathcal{T}_{2,v}$, according to Lemma 2, we have

$$
\begin{aligned}
\mathcal{T}_{2,v} &\leq \frac{L_v}{2}\Big(\frac{16\eta_v^2 K_v^2}{m}\sum_{i=1}^m \mathbb{E}\|\nabla_v F_i(u_i^t, v_i^t)\|^2 + 8\eta_v^2 K_v^2 \sigma_v^2\Big) \\
&\leq 8L_v \eta_v^2 K_v^2 \mathbb{E}[\Delta_v^t] + 4L_v \eta_v^2 K_v^2 \sigma_v^2.
\end{aligned}
\tag{30}
$$

For $\mathcal{T}_{3,v}$, according to Eq. (25), we have

$$
\frac{L_{vu}^2}{2m}\sum_{i=1}^m \mathbb{E}\|\bar{u}^t - u_i^t\|^2 \leq \frac{L_{vu}^2}{(1-\lambda)^2}\Big[3\eta_u^2 K_u L_u^2 \rho^2 + 9\eta_u^2 K_u^2\big(\sigma_u^2 + \delta^2 + 2\mathbb{E}[\Delta_{\bar{u}}^t]\big)\Big].
\tag{31}
$$

After that, summing Eq. (29), (30), and (31), we have

$$
\begin{aligned}
\mathbb{E}\Big[F\big(\bar{u}^t, V^{t+1}\big) - F\big(\bar{u}^t, V^t\big)\Big] &\leq \Big(-\eta_v K_v + 8\eta_v^2 K_v^2(L_v + 1)\Big)\mathbb{E}[\Delta_v^t] + 4\eta_v^2 K_v^2 \sigma_v^2(L_v + 1) \\
&\quad + \frac{L_{vu}^2}{(1-\lambda)^2}\Big[3\eta_u^2 K_u L_u^2 \rho^2 + 9\eta_u^2 K_u^2\big(\sigma_u^2 + \delta^2 + 2\mathbb{E}[\Delta_{\bar{u}}^t]\big)\Big].
\end{aligned}
\tag{32}
$$

**Obtaining the Final Convergence Bound.**

$$\mathbb{E}\Big[F\left(\bar{u}^{t+1}, V^{t+1}\right) - F\left(\bar{u}^t, V^t\right)\Big] = \mathbb{E}\Big[F\left(\bar{u}^t, V^{t+1}\right) - F\left(\bar{u}^t, V^t\right) + F\left(\bar{u}^{t+1}, V^{t+1}\right) - F\left(\bar{u}^t, V^{t+1}\right)\Big]$$

$$\leq \left(\frac{\eta_u}{2} - \eta_u K_u + \frac{54\eta_u^3 K_u^3 L_u^2}{(1-\lambda)^2}\right)\mathbb{E}[\Delta_{\bar{u}}^t] + \frac{5}{2}\eta_u^2 K_u L_u\left(L_u^2 \rho^2 + \sigma_u^2 + \delta^2\right)$$

$$+ \frac{3\eta_u K_u L_u^2}{2(1-\lambda)^2}\left(6\eta_u^2 K_u L_u^2 \rho^2 + 18\eta_u^2 K_u^2(\sigma_u^2 + \delta^2)\right)$$

$$+ \left(-\eta_v K_v + 8\eta_v^2 K_v^2(L_v + 1)\right)\mathbb{E}[\Delta_v^t] + 4\eta_v^2 K_v^2 \sigma_v^2(L_v + 1)$$

$$+ \frac{L_{vu}^2}{(1-\lambda)^2}\left[3\eta_u^2 K_u L_u^2 \rho^2 + 9\eta_u^2 K_u^2\left(\sigma_u^2 + \delta^2 + 2\mathbb{E}[\Delta_{\bar{u}}^t]\right)\right]. \tag{33}$$

Summing from $t = 1$ to $T$, assume the local learning rates satisfy $\eta_u = \mathcal{O}(1/L_u K_u \sqrt{T}), \eta_v = \mathcal{O}(1/L_v K_v \sqrt{T})$, $F^*$ is denoted as the minimal value of $F$, i.e., $F(\bar{u}, V) \geq F^*$ for all $\bar{u} \in \mathbb{R}^d$, and $V = (v_1, \ldots, v_m) \in \mathbb{R}^{d_1 + \ldots + d_m}$. We can generate

$$\frac{1}{T}\sum_{i=1}^T \left(\frac{1}{L_u}\mathbb{E}[\Delta_{\bar{u}}^t] + \frac{1}{L_v}\mathbb{E}[\Delta_v^t]\right) \leq \mathcal{O}\Big(\frac{F(\bar{u}^1, V^1) - F^*}{\sqrt{T}} + \frac{\sigma_v^2(L_v + 1)}{L_v^2\sqrt{T}} + \frac{L_u^2\rho^2 + \sigma_u^2 + \delta^2}{L_u K_u \sqrt{T}}$$

$$+ \frac{L_{vu}^2}{(1-\lambda)^2\sqrt{T}}\Big(\frac{\rho^2}{K_u} + \frac{\sigma_u^2 + \delta^2}{L_u^2}\Big) + \frac{L_u}{(1-\lambda)^2 T}\Big(\frac{\rho^2}{K_u} + \frac{\sigma_u^2 + \delta^2}{L_u^2}\Big)\Big). \tag{34}$$

Assume that

$$\sigma^2 = \frac{\rho^2}{K_u} + \frac{\sigma_u^2 + \delta^2}{L_u^2}, \quad \sigma_4^2 = \frac{\sigma_v^2(L_v + 1)}{L_v^2} + \frac{L_u^2\rho^2 + \sigma_u^2 + \delta^2}{L_u K_u},$$

Then, we have the final convergence bound:

$$\frac{1}{T}\sum_{i=1}^T \left(\frac{1}{L_u}\mathbb{E}[\Delta_{\bar{u}}^t] + \frac{1}{L_v}\mathbb{E}[\Delta_v^t]\right) \leq \mathcal{O}\Big(\frac{F(\bar{u}^1, V^1) - F^*}{\sqrt{T}} + \frac{\sigma^2 L_{vu}^2}{(1-\lambda)^2\sqrt{T}} + \frac{\sigma_4^2}{\sqrt{T}} + \frac{\sigma^2 L_u}{(1-\lambda)^2 T}\Big). \tag{35}$$

Furthermore, When the perturbation amplitude $\rho$ is proportional to the learning rate, e.g., $\rho = \mathcal{O}(1/\sqrt{T})$, the sequence of outputs $\Delta_{\bar{u}}^t$ and $\Delta_v^t$ generated by Alg. 1, we have:

$$\mathcal{O}\left(\sigma^2\right) = \mathcal{O}\Big(\frac{\rho^2}{K_u} + \frac{\sigma_u^2 + \delta^2}{L_u^2}\Big) = \mathcal{O}\Big(\frac{1}{K_u T} + \frac{\sigma_u^2 + \delta^2}{L_u^2}\Big). \tag{36}$$

