# OpenReview forum: "Enhancing Personal Decentralized Federated Learning through Model Decoupling"
_ICLR.cc/2024/Conference — Submitted to ICLR 2024_

### Official Review · Reviewer_7fx1 · 2023-10-26

**Soundness:** 3 good
**Presentation:** 4 excellent
**Contribution:** 3 good
**Rating:** 6
**Confidence:** 3

**Summary:**

This work proposes a personalized federated learning (PFL) framework called DFEedMDC, which pursues robust communication and better model performance with a convergence guarantee. Besides, to promote the shared parameters aggregation process, the authors propose DFedSMDC via integrating the local Sharpness Aware Minimization (SAM) optimizer to update the shared parameters.

**Strengths:**

This work designs personalized local models and training schemes for decentralized federated learning. The authors present theoretical analyses for the convergence, which shows the negative influence of the statistical heterogeneity and the communication topology. Extensive experiments are conducted to evaluate the effectiveness of the proposed methods.

**Weaknesses:**

What do the "Grid" and "Exp" represent in Fig. 3? It would be easier for the readers to understand different communication topologies by visualizing them in the main test or in the Appendix.

In light of Theorem 1 and Theorem 2, the communication topology (i.e., the eigenvalue $\lambda$) has an impact on the DFedMDC and DFedSMDC methods. The reviewer suggests the authors report the $\lambda$ values of different communication topologies in Fig. 3 and discuss the influence of $\lambda$ on the test accuracy.

**Questions:**

The proposed DFedSMDC method, a variant of DFedMDC, achieves better performance by integrating the SAM optimizer into the local iteration update of shared parameters. The reviewer is curious if the incorporation of this optimizer could similarly enhance the performance of other baseline methods.

---

> ### Author Response · Authors · 2023-11-22
> **Reply to Reviewer 7fx1.**
>
> We greatly appreciate your suggestions and comments, which will make our work better than this version. Thanks for your positive comments, and we address the weaknesses and minor comments below.
>
> ## W1:  What do the "Grid" and "Exp" represent?
> Thank you for your suggestion and we added the visualization of topologies in Appendix B.3 in the revision.  The explanations of Grid and Exp are as follows:
> - __Grid__: Grid topology refers to a network configuration where clients are arranged in a two-dimensional grid-like pattern. That means clients are organized in rows and columns, forming a rectangular or square grid structure. In a 100-client network, the exponential neighbors of client 10 are client 11(10+1), client 9(10-1), client 19(10+9), client 1(10-9).
> - __Exp__: Exponential(Exp) topology refers to a network configuration where clients are arranged in an exponential manner. In a 100-client network, the exponential neighbors of client 10 are client 11(10+$2^0$), client 12(10+$2^1$), client 14(10+$2^2$), client 18(10+$2^3$), client 26(10+$2^4$), client 42(10+$2^5$), client 74(10+$2^6$).
>
> ## W2:  The value and the influence of $\lambda$ in Fig.3.
> Thank you for your suggestion and we have added more analysis of the spectral gap $(1-\lambda)$ of different communication topologies to the revision.  From the relationship between the spectral gap $(1-\lambda)$ and the participation number of clients $m$ ($m=100$ in our experiment), as follows, we can see that the convergence bounds of different topologies are ranked as follows: Fully-connected > Exponential > Grid > Ring, which is consistent with our experiment results.
> Graph Topology | Spectral Gap $1-\lambda$
> --- | :--:|
> Fully-connected|1
> Disconnected|0
> Ring| $\approx 16\pi^2/3m^2$
> Grid| $\mathcal{O}(1/(mlog_2(m)))$
> Exponential Graph|$2/(1 + log_2(m))$
>
> ## Q1:  Incorporation of SAM optimizer with other baseline methods.
> We add more experiments about the incorporation of SAM optimizer with other baseline methods on CIFAR-10 as follows.
> Algorithm| FedAvg| FedPer | FedRep  | FedBABU | FedRod |Ditto |  DFedAvgM| Dis-PFL |DFedMDC
> --- | :--:| :--: | :--: | :--:| :--: | :--: | :--: | :--: | :--:
> Dir0.3|79.66|84.06|84.50|83.26|85.68|73.51|82.60 |82.71|86.50
> Dir0.3+SAM| 80.02 |86.81|87.19|85.61|85.33|71.07|77.36 |82.66|88.32
> Pat2|85.04|90.94|91.16|91.17|90.10|84.96|90.72|88.19|91.26
> Pat2+SAM|84.99|91.73|91.80|91.67|89.52|83.36|90.14|87.89|92.21
>
> From the experimental results, SAM could similarly enhance the performance of the partial model personalized methods, but whether it could enhance the full model personalized methods is doubtful. For the partial model personalized methods FedPer, FedRep, FedBABU and DFedMDC, SAM adds proper perturbation in the direction of the shared models’ gradient to make the shared part more robust and flat, which will benefit the average progress for each client. For the full model personalized methods FedAvg, FedRod, Ditto and DFedAvgM, pursuing the full model flatteness will decrease the classifier’s adaptation to local distribution.

---

> > ### Comment · Reviewer_7fx1 · 2023-11-23
> >
> > Thank you for the responses. After careful consideration, I decided to keep my initial score.

---

### Official Review · Reviewer_z1Hn · 2023-10-30

**Soundness:** 2 fair
**Presentation:** 2 fair
**Contribution:** 1 poor
**Rating:** 3
**Confidence:** 3

**Summary:**

This paper studies the personalized federated learning problem under fully decentralized setting. The framework of the considered personalized learning is the commonly used model decoupling with a globally shared model and personalized local models. DFedSMDC, an algorithm via integrating the local Sharpness Aware Minimization (SAM) optimizer to update the shared parameters, is proposed. Theoretical convergence results and numerical experimental results are both presented.

**Strengths:**

This paper studies the personalized federated learning problem under fully decentralized setting, and proposed DFedSMDC, an algorithm via integrating the local Sharpness Aware Minimization (SAM) optimizer to update the shared parameters.

**Weaknesses:**

1. The reviewer is quite doubt about the final results as shown in Theorem 1 and Theorem 2. I’ve checked the theoretical proof in the appendix and do not find the exact expressions for the final convergence results, but only the $\mathcal{O}$ expression. The first questionable part is that the right-hand side of Eqs. (3)-(4) will goes to 0 as the number of rounds $T$ goes to infinity, while in reality, this is not true for non-i.i.d scenarios. There will exists some constant terms related with heterogeneity that are irrelevant to $T$. Please explain this.

2. The second part that may not be true in the theoretical results is that the convergence speed is monotonically related with the spectral gap $\lambda$. If this is true, it solves the challenging topology design problem of decentralized federated learning, since a fully-connected topology is the optimal topology according to the theoretical results in this paper. There is no discussion about this point in current manuscript and this leads to a doubtful result.

3. Why is the convergence results not related with the number of workers? This is also a weird part.

4. Why Theorem 1 is related with the cross Lipschitz constant $L_{vu}$, and Theorem 2 is related with $L_{vu}$? How about $L_{uv}$?

5. The results in Fig.3 are questionable according to the second comment. The reviewer is not sure if a fully-connected topology is the best.

6. What is the meaning of Fig. 4? Are multiple local epochs good or bad? How is it related with the theoretical results?

**Questions:**

See the weakness above.

---

> ### Author Response · Authors · 2023-11-22
> **Reply to Reviewer z1Hn(1).**
>
> Thanks for your comments and we reply to them below. Please reconsider the contributions of our work.
> ## W1: The doubts about Theorem 1 and Theorem 2.
> - __Only the $\mathcal{O}$ expression.__ $\mathcal{O}$ is the final results after assuming the learning rates $\eta_u$ and $\eta_v$. Similar results can be found in [1-3]. Due to the limited space, the initial convergence analysis has been presented in Formula (19) and Formula (33).
> - __The right-hand side of Eqs. (3)-(4) will goes to 0 as the number of rounds $T$ goes to infinity.__
>     - The gradient that goes to 0 is one of the optimality conditions in non-convex optimization.  As an optimization problem, we consider minimizing problem: $\min _{x \in \mathbb{R}^d} f(x)$ and find the global minima $x^*$, that satisfies $f(x) \geq f(x^*)$. In non-convex optimization, the minima can be defined by  $\nabla f\left(x^{\star}\right)=0$.  So we prove the convergence from the gradients going to 0, which satisfies $\| \nabla f(x^T)\| \leq \epsilon$. More details about non-convex optimization can be referred to [4,5].
>     - We assume that $\eta_u= \mathcal{O}(1/L_uK_u\sqrt{T})$ and $\eta_v= \mathcal{O}(1/L_vK_v\sqrt{T})$, which means that the learning rate $\eta_u$ and $\eta_v$ tend to 0 as the number of rounds $T$ goes to infinity, and the gradients eventually tend to 0 correspondently. We deduce that the reviewer wants to say that the proposed methods will converge to a variance under the constant learning rate. Actually, our methods converge to a variance domain with the sublinear order convergence speed.
>     -  Moreover, in a non-iid scenario, each client will achieve convergence and their gradient will approximate to 0 after sufficient large communication rounds. However, this does not mean that all clients have the same solution. For PFL, every client will own its unique solution $(u^*,v_i^*)$, the $v_i^*$ of which is different from each other.
>
> ## W2: The convergence speed is monotonically related to the spectral gap.
> - __The convergence speed is monotonically related to the spectral gap.__ The convergence speed is related to the statistical heterogeneity $\delta$, the smoothness $L_u$, $L_v$, $L_{vu}$ of loss functions, the local epochs $K_u$ and the communication topology (1 − $\lambda$) in the theoretical results. Due to the limited space, we only present the final results in Formula (3) and Formula (4). Please check the details in Formula (20-21) and Formula (34-36).
> - __Whether Fully-connected topology is the optimal topology according to the theoretical results.__
>     - Theoretically, Fully-connected topology is the optimal topology among the comparison topologies and the comparison of the spectral gap of each topology is as follows ($m$ refers to the partition number of clients):
>         Graph Topology | Spectral Gap $1-\lambda$
>         --- | :--:|
>         Fully-connected|1
>         Disconnected|0
>         Ring| $\approx 16\pi^2/3m^2$
>         Grid| $\mathcal{O}(1/(mlog_2(m)))$
>         Exponential|$2/(1 + log_2(m))$
>
>         It is clear to see that the convergence bounds of different topologies are ranked as follows: Fully-connected > Exponential > Grid > Ring. __But it contains more communication cost compared with other topologies.__ So it is a trade-off between the convergence and communication cost in the real world. We have added the comparison to the revision.
>     - Empirically, we conduct experiments on different topologies (i.e. Ring, Grid, Exponential and Fully-connected）in Fig 3, which suggests the personalized performance of different topologies is ranked as follows: Fully-connected > Exponential > Grid > Ring.
>
> ## W3: The convergence results are not related to the number of workers.
> Refer to the table above, the effects of the participation workers $m$ have been involved in the spectral gap $(1-\lambda)$ of communication topologies. We have added it to the revision.
>
> [1] Decentralized federated averaging, ICML2022.
>
> [2] Improving the Model Consistency of Decentralized Federated Learning, ICML2023.
>
> [3]Federated Learning with Partial Model Personalization, ICML2022.
>
> [4] Non-convex Optimization for Machine Learning.
>
> [5] First-order methods in optimization.

---

> > ### Author Response · Authors · 2023-11-22
> > **Reply to Reviewer z1Hn(2).**
> >
> > ## W4: Theorem 1 and Theorem 2 are all related to the cross Lipschitz constant $L_{vu}$.
> > - First, based on the Lipschitz Smooth assumption, we define $\nabla_u F_i(u_i,v_i)$ is $L_u$--Lipschitz with respect to $u_i$ and $L_{uv}$--Lipschitz with respect to $v_i$ in Assumption 1, which means $\|\nabla_u F_i(u_i,V)-\nabla_u F_i(u_j,V)\| \leq L_u \|u_i - u_j\|$ and $\|\nabla_u F_i(u,v_i)-\nabla_u F_i(u,v_j) \| \leq L_{uv} \|v_i-v_j\|$.
> >     The same for $\nabla_v F_i(u_i,v_i)$ is $L_v$--Lipschitz with respect to $v_i$ and $L_{vu}$--Lipschitz with respect to $u_i$. More details about Lipschitz Smooth assumption can be found in [6].
> >
> >  - Then, we conduct
> > $\frac{1}{m}\sum_{i=1}^{m} E \big< \nabla_v F_i(\bar{u}^{t}, v_{i}^{t}) - \nabla_{v} F_{i} \left( u_{i}^{t}, v_{i}^{t} \right), v_{i}^{t+1} - v_{i}^{t} \big> \leq \frac{1}{2m} \sum_{i=1}^{m} \||\nabla_{v} F_{i} \left(\bar{u}^{t}, v_{i}^{t}\right)  - \nabla_{v} F_{i}\left(u_{i}^{t}, v_{i}^{t}\right)\||^2 +  \frac{1}{2m}\sum_{i=1}^m E \||v_{i}^{t+1} - v_{i}^{t}\||^2  \leq \frac{L_{vu}^2}{2m}\sum_{i=1}^m E \||\bar{u}^t-u_{i}^{t}\||^2 + \frac{1}{2m}\sum_{i=1}^m E \||v_{i}^{t+1} - v_{i}^{t}\||^2.$
> > in the second line in Formula (15).
> >
> >  - The theoretical analysis progress is consistence with the algorithm, which first updates the personalized parts $v_i$ and then the shared parts $u_i$. So Theorem 2 and Theorem 1 are associated with $L_{vu}$ not $L_{uv}$.
> >
> >
> > ## W5: Is a fully-connected topology is the best?
> > Fully-connected topology achieves the best performance in the comparisons under different topologies. But it needs more communication costs for each client. So it is a trade-off of the test performance and communication cost to choose the topology.
> >
> > ## W6: The meaning of Fig. 4.
> > - Local epochs $K_v$ for the personal parameters $v_i$ is an important hyperparameter in our algorithm, which will significantly influence the final experiment results. From the comparison of FedBABU($K_v$=0) and FedRep($K_v$=10), we deduce $K_v$ is an important hyperparameter in the alternately partial personalization. $K_v$ represents the trade-off of the contribution of the shared representation and personalized classifiers to the convergence. Bigger $K_v$ means the local convergence relied more on the fitting ability of the personalized classifiers, while smaller $K_v$ means the local convergence relied more on the generalization of the shared representation. This point has never been discussed before in the partial personalization works, which is one of our surprising findings as well.
> > - Theoretically, multiple local epochs speed up the convergence but may be prone to over-fitting. This will decrease the generalization accuracy and it is a common phenomenon in FL. Similar to it is the large batch-size training. The convergence will be accelerated with large batch sizes, but the generalization will become worse. Multiple local epochs good or bad are related to the data distribution in our experiments and similar performance appears in [7] as well.
> > - $K_v$ has been cancelled out from the assumption $\eta_v= \mathcal{O}(1/L_vK_v\sqrt(T))$ in Formula (19-20).
> >
> >
> > [6] Nonlinear Systems 3rd.
> >
> > [7]On Large-Cohort Training for Federated Learning, Neurips 2021.

---

> > > ### Comment · Reviewer_z1Hn · 2023-11-23
> > > **Thank you for your response.**
> > >
> > > I read the authors' response and will keep my original score.

---

> > > > ### Author Response · Authors · 2023-11-23
> > > > **Thank you for your feedback.**
> > > >
> > > > Thanks for your reply. We acknowledge your effort in reviewing our paper and providing valuable feedback.
> > > >
> > > > If you have checked our response, can you enumerate where else we can improve?
> > > >
> > > > Thanks

---

### Official Review · Reviewer_iEM9 · 2023-10-30

**Soundness:** 2 fair
**Presentation:** 2 fair
**Contribution:** 2 fair
**Rating:** 5
**Confidence:** 2

**Summary:**

In this paper, the authors present an innovative framework known as DFedMDC, which leverages model decoupling to address these issues and aims to provide robust communication and superior model performance while guaranteeing convergence. DFedMDC achieves this by personalizing the "right" components within modern deep models through alternate updates of shared and personal parameters, facilitating the training of partially personalized models in a peer-to-peer manner. To enhance the shared parameters aggregation process, the authors introduce DFedSMDC, which incorporates the local Sharpness Aware Minimization (SAM) optimizer to update shared parameters. SAM optimizer introduces proper perturbations in the gradient direction to mitigate inconsistencies in the shared model across clients.

The paper provides a thorough theoretical foundation, offering a convergence analysis of both algorithms in a general non-convex setting with partial personalization and SAM optimizer for the shared model.

**Strengths:**

1. The paper is well-written and exhibits a high degree of clarity, making it accessible and easy to comprehend.
2. The paper's strength is further underscored by its meticulous convergence analysis, enhancing its overall robustness.
3. The paper substantiates its claims with an exhaustive array of experimental results, effectively confirming the effectiveness of the proposed method.

**Weaknesses:**

1. A significant concern revolves around the novelty of the proposed method. The concept of model decoupling in personalized federated learning [1] and the application of Sharpness Aware Minimization (SAM) [2] to address model inconsistencies in decentralized federated learning have both been extensively explored in the literature. As such, the proposed method may appear to be a fusion of existing ideas (resembling an 'A+B' approach). It is essential for the authors to underscore their distinctive contributions in a more prominent manner.

2. In terms of experimental baselines, it is recommended that the authors include the most recent decentralized federated learning method ([2]) for a comprehensive comparison. This will enhance the paper's completeness and relevance in the context of the current state of the field.

3. Regarding the convergence analysis, it would be valuable to incorporate a discussion that compares the proposed method's convergence rate with the state-of-the-art (SOTA) approaches.

[1] Exploiting Shared Representations for Personalized Federated Learning
[2] Improving the Model Consistency of Decentralized Federated Learning

**Questions:**

See weaknesses section above.

---

> ### Author Response · Authors · 2023-11-22
> **Reply to Reviewer iEM9.**
>
> Thanks for your comments and we reply to them below. Please reconsider the contributions of our work.
>
> ## W1: The novelty of our work.
> We give more details about our background, motivations and contributions. Please check them in the top official comment box.
>
> ## W2: Add the most recent decentralized federated learning method for a comprehensive comparison.
> Thanks for your suggestion. We provide a comparison with DFedSAM in the following and we have added this comparison in the revision.
>
> | **Dataset**   |    | **Cifar-10** |  |   |    | **Cifar-100** | |   |
> |:-------------:|:-------|:------------:|:-----:|:-----:|:-------:|:-------------:|:-----|:------|
> | **Algorithm** | Dir-0.1 | Dir-0.3      | Pat-2 | Pat-5 | Dir-0.1 | Dir-0.3       | Pat-5 | Pat-10 |
> | **DFedSAM**|84.96| 77.36|90.14|83.05|58.21|47.80|74.25|67.34|
> | **DFedAvgM**  | 87.39   | 82.60        | 90.72 | 84.69 | 59.76   | 54.98         | 76.70 | 71.08  |
> | **DFedMDC**| 88.85   | 86.50        | 91.26 | 86.85 | 66.26   | 57.66         | 78.78 | 72.19  |
> | **DFedSMDC**  | 91.08 | 87.67|92.20|88.34|67.03|58.73|80.82|74.50|
>
> From the comparison between DFedAvgM and DFedSAM we can see that adding the gradient perturbation to the full model will increase the model consistency but hurt the personalized performance. On the contrary, adding the perturbation to the shared representation parts in DFedSMDC outperforms DFedMDC clearly, which shows that both improving the shared representation parts consistency and keeping the private classifiers locally are both the key points to facilitate the personalized performance in decentralized PFL.
> ## W3: compares the proposed method's convergence rate with the state-of-the-art (SOTA) approaches.
> Compared with the SOTA bounds $\mathcal{O}\Big(\frac{1}{\sqrt{T}}+\frac{\sigma_l^2 + K \sigma_g^2}{K \sqrt{T}}+\frac{\sigma_l^2 + K\sigma_g^2+ KB^2}{K(1-\lambda)^2T^{3/2}}\Big)$  ( is the upper bound of the gradient) of existing work DFedAvg and $\mathcal{O}\Big(\frac{1}{\sqrt{KT}}+\frac{K(\sigma_g^2+\sigma_l^2)}{T}+\frac{\sigma_g^2+\sigma_l^2}{K^{1/2}(1-\lambda)^2T^{3/2}}\Big)$ of DFedSAM in decentralized works, our algorithms reflect the impact of the L-smoothness and the gradient variance of the shared model $u$ and personalized model $v$ on convergence rate. On the other hand, compared with the SOTA bound of FedAlt in personalized works, our algorithms reflect the impact of the communication topology $\lambda$ (the value of  that increases when connectivity is more sparse, which has been studied by existing work [1].)
>
> Therefore, the main contribution of our convergence analysis is the first to analyze the impact of model decoupling and the decentralized networks on the convergence rate instead of delivering the SOTA bound compared with others. Meanwhile, this theoretical analysis can be mutually verified with experimental results.
>
> [1] Topology-aware Generalization of Decentralized SGD. ICML2023.

---

> ### Author Response · Authors · 2023-11-23
> **Last Time Reminder & Sincerely Reply to Reviewer iEM9.**
>
> Dear Reviewer iEM9,
>
> We sincerely appreciate the time and effort you have invested in reviewing our work. As the deadline for the discussion period is approaching, we kindly request that you inform us of any remaining questions you may have.
>
> We are confident that our response has adequately addressed your concerns, and we would be grateful for your feedback. Should you require further clarification, we would be delighted to answer any additional questions and provide more information.
>
> Best wishes,
>
> The Authors

---

### Official Review · Reviewer_A4M3 · 2023-10-31

**Soundness:** 2 fair
**Presentation:** 2 fair
**Contribution:** 2 fair
**Rating:** 5
**Confidence:** 3

**Summary:**

This paper proposes interesting methods DFedMDC and DFedSMDC for PFL, which simultaneously guarantee robust communication and better model performance with convergence guarantee via adopting decentralized partial model personalization based on model decoupling.

**Strengths:**

1. The study of personalized FL on decentralized FL is meaningful.
2. The experiments demonstrate that the proposed method is useful.

**Weaknesses:**

1. The proposed algorithm seems trivial and common in PFL. It seems its idea is the adoption of the method in DFL. Can you clarify what is the main novelty of this method?
2. Why introduce the SAM? It is unclear about the advantage of introducing this optimizer. Can you elaborate on it intuitively and theoretically?
3. In the theorem, why is it $V^{t+1}$, instead of $V^{t}$, and what does it mean?
4. The experiment results are a bit weird, in Table 1. Why do all baselines achieve better performance under larger heterogeneity? As I know, larger heterogeneity will usually lead to worse performance [1].
5. Regarding ``The test performance will get a great margin with the participation of clients decreasing’’: What will happen when the client number is less than 10, even 1? Does it mean no collaboration is the best?

[1]Karimireddy S P, Kale S, Mohri M, et al. Scaffold: Stochastic controlled averaging for federated learning[C]//International conference on machine learning. PMLR, 2020: 5132-5143.

Minors:

1.	It seems the hyperparameters of the proposed methods are finetuned (like $rho$ and local epoch for the personal part). Are the baselines’ results well finetuned? What's the used hyperparameter for baselines?
2.	What is the definition of $\sigma$ in Theorem 2?

**Questions:**

1. Could you give more explanation on Theorem 2? What is the difference/advantage compared with Theorem 1 as you introduce SAM into shared parameters?
2. Can you provide baseline results with more hyperparameter settings?
3. Could the authors provide more details about the experiment settings?

---

> ### Author Response · Authors · 2023-11-22
> **Reply to Reviewer A4M3(1).**
>
> Thanks for your comments and we reply to them below. Please reconsider the contributions of our work.
> ## W1: Clarify the main novelty of this method.
> We clarify more details about our background, motivation and contribution. Please check them in the top official comment.
> ## W2: Why introduce the SAM?
> - Intuitively, SAM adds proper perturbation in the gradient direction to make the shared part become more robust and flatter, which will benefit the model average between clients.
> - Theoretically, we give the effect on bound after adding the operation of SAM in Remark 2, where assuming $\rho = \mathcal{O}(\frac{1}{\sqrt{T}})$. Due to the limited space, we have  presented the bound before assuming $\rho = \mathcal{O}(\frac{1}{\sqrt{T}})$ in Formula (34) in Appendix C.2.2 on Page 26 as follows:
>     - before using $\rho = \mathcal{O}(\frac{1}{\sqrt{T}})$:
>     $\begin{aligned} &\frac{1}{T} \sum_{i=1}^T\left(\frac{1}{L_u} \mathbb{E}\left[\Delta_{\bar{u}}^t\right]+\frac{1}{L_v} \mathbb{E}\left[\Delta_v^t\right]\right)  \leq \mathcal{O}\left(\frac{F\left(\bar{u}^1, V^1\right)-F^*}{\sqrt{T}}+\frac{\sigma_v^2\left(L_v+1\right)}{L_v^2 \sqrt{T}}+\frac{L_u^2 \rho^2+\sigma_u^2+\delta^2}{L_u K_u \sqrt{T}}\right.  \left.+\frac{L_{v u}^2}{(1-\lambda)^2 \sqrt{T}}\left(\frac{\rho^2}{K_u}+\frac{\sigma_u^2+\delta^2}{L_u^2}\right)+\frac{L_u}{(1-\lambda)^2 T}\left(\frac{\rho^2}{K_u}+\frac{\sigma_u^2+\delta^2}{L_u^2}\right)\right) .\end{aligned}$
>     - after using $\rho = \mathcal{O}(\frac{1}{\sqrt{T}})$:
>     $\begin{aligned} &\frac{1}{T} \sum_{i=1}^T\left(\frac{1}{L_u} \mathbb{E}\left[\Delta_{\bar{u}}^t\right]+\frac{1}{L_v} \mathbb{E}\left[\Delta_v^t\right]\right)  \leq \mathcal{O}\left(\frac{F\left(\bar{u}^1, V^1\right)-F^*}{\sqrt{T}}
>     +\frac{\sigma_v^2\left(L_v+1\right)}{L_v^2 \sqrt{T}}
>     +\frac{\sigma_u^2+\delta^2}{L_u K_u \sqrt{T}}\right. \left.+\frac{L_{v u}^2(\sigma_u^2+\delta^2)}{L_u^2(1-\lambda)^2 \sqrt{T}}
>     +\frac{\sigma_u^2+\delta^2}{L_u(1-\lambda)^2 T}+ \frac{L_u}{K_u\sqrt{T^3}}+\frac{L_{vu}^2}{K_u(1-\lambda)^2\sqrt{T^3}}+\frac{L_u}{K_u(1-\lambda)^2T^2}\right) .\end{aligned}$
> - Empirically, the comparison between DFedSMDC and DFedMDC in different heterogeneity and different topologies can significantly demonstrate the effectiveness in enhancing the consistency of the shared models in decentralized PFL.
> ## W3: $V^{t}$ not $V^{t+1}$ in the theorem.
> Thanks for your suggestion and we have polished it in the revision paper.
> ## W4: "Why do all baselines achieve better performance under larger heterogeneity? "
> What we focus on are the PFL problems, not the FL problems. In PFL tasks, the higher heterogeneity of data distribution means it owns fewer classes of data locally, which makes the classification task easier and clients will achieve better performance. For example, in the Pathological-2 setting, the local task is a binary classification task, which is easier than the five classification tasks in the Pathological-5 setting, so the average test performance in Pathological-2 is better than that in Pathological-5. The same phenomenon can be seen in most PFL works [1, 2, 3, 4].
>
> ## W5:  "The test performance will get a great margin with the participation of clients decreasing.’’
> We have updated more experiments when the client number is 10 and 5 in Fig 5 in the revision paper. The test performance still improves with the participation of clients decreasing. And we clarify our opinion theoretically and empirically as follows:
> - Theoretically, the convergence bound is related to the topology, which is associated with the participation of clients. The relationship between the topology and the participation clients $m$ is as below:
>     Graph Topology | Spectral Gap 1-$\lambda$
>     --- | :--:|
>     Fully-connected|1
>     Disconnected|0
>     Ring| $\approx 16\pi^2/3m^2$
>     Grid| $\mathcal{O}(1/(mlog_2(m)))$
>     Exponential Graph|$2/(1 + log_2(m))$
>
>     When the participation clients $m$ decrease, the spectral gap increases and the convergence bound is tighter.
> - Empirically, with the participation of clients decreasing, the number of local data increases. More data will help to achieve a better performance.
> - When the client number is 1, the problem will become a classical ML problem based on a centralized data assumption, not an FL problem based on a decentralized data setting. We aim to collaboratively train the personalized models without transmitting data.
>
> [1]On bridging generic and personalized federated learning for image classification. ICLR2022.
>
> [2]Fedbabu: Towards enhanced representation for federated image classification. ICLR2022.
>
> [3]Personalized federated learning with feature alignment and classifier collaboration. ICLR2023.
>
> [4]FedProto: federated prototype learning across heterogeneous clients AAAI2022.

---

> ### Author Response · Authors · 2023-11-22
> **Reply to Reviewer A4M3(2).**
>
> ## M1: What's the used hyperparameter for baselines? Are the baselines’ results well fine-tuned?
> The baselines’ results have been well-finetuned and the hyperparameter of the baselines can be found in Appendix B.3 (MORE DETAILS ABOUT BASELINES).  We show the finetuned processing as follows.
> - __Setting__: For all methods, we keep the experiment setting as the same as follows. We perform 500 communication rounds with 100 clients. The client sampling radio is 0.1 in CFL, while each client communicates with 10 neighbors in DFL. The batch size is 128. The learning rate is 0.1 with 0.99 exponential decay. The local epochs are set to 5 for full model personalized methods and the shared models of the partial model personalized methods. The local optimization is SGD with momentum 0.9 and 5e-4 weighted decay.
> - __Finetuning__: For FedAvg, FedPer, FedBABU, Fed-RoD, DFedAvgM, there is no more hyperparameters. We show the finetuning progress for FedRep, FedSAM, Ditto, Dis-PFL on CIFAR-10 as follow:
>     - __FedRep__: We finetune the local epochs for the personalized parts $K_v$ and set $K_v=10$ in our experiments.
>          $K_v$ | 1|4|7|10|15|
>         --- | :--:|:--:|:--:|:--:|:--:|
>         Dir-0.3 |88.50|90.71|90.81|91.09|90.79
>         Pat-2 |80.85|83.93|84.17|84.50|84.27
>
>         Theoretically, multiple local epochs will speed up convergence but may be prone to over-fitting. It is a trade-off between convergence and generalization. The difference of the best epochs for the personal part demonstrates the difference between CFL and DFL.
>     - __FedSAM__: The extra hyperparameter in FedSAM is the perturbation radius $\rho$ and we set $\rho = 0.7$ in our experiments.
>         $\rho$ | 0.01|0.1|0.3|0.5|0.7|0.9
>         --- | :--:|:--:|:--:|:--:|:--:|:--:|
>         Dir-0.3 |78.78|79.08|79.20|79.61|80.02|79.52
>         Pat-2 |83.26|83.69|84.11|84.57|84.99|83.48
>
>         With a larger $\rho$, the flatter models that benefit the aggregation average can increase the global generalization. But for PFL, it is a trade-off between the personalization and the generalization.
>     - __Ditto__: The extra hyperparameter in Ditto is the  interpolated coefficient $\lambda$ and we set $\lambda = 0.75$ in our experiments.
>         $\lambda$ | 0.1|0.4|0.75|1.25|2|
>         --- | :--:|:--:|:--:|:--:|:--:|
>         Dir-0.3 |64.77|70.63|73.51|72.73|69.50
>         Pat-2 |80.10|83.64|84.96|84.78|76.71
>
>        $\lambda$ offers a trade-off between the personalization and the generalization. When $\lambda$ is set to 0, Ditto is reduced to training local models; as $\lambda$ grows large, it recovers the global model objective.
>     - __Dis-PFL__:We finetuned the sparsity ratio in Dis-PFL and set it as 0.5 in our experiments.
>         Sparsity Ratio | 0.2|0.4|0.5|0.6|0.8|
>         --- | :--:|:--:|:--:|:--:|:--:|
>         Dir-0.3 |80.17|82.63|82.72|82.56|81.94
>         Pat-2 |87.77|88.07|88.19|87.99|87.82
>
>       The sparsity ratio is a trade-off between generalization and personalization. A higher sparsity ratio may bring more generalization benefits when the sparsity is small. But with the sparsity ratio increasing, less information exchange will lead to performance degradation.
> ## M2: What is the definition of $\sigma$ in Theorem 2?
> $\sigma$ is a variate associate with the gradient perturbation $\rho$ in SAM. We define $\sigma^2 =\frac{\rho^2}{K_u}+\frac{\sigma^2_u+\delta^2}{L_u^2}$ in Formula (24-26). And when assuming $\rho = \mathcal{O}(\frac{1}{\sqrt{T}})$, $\mathcal{O}(\sigma^2)= \mathcal{O}(\frac{1}{K_uT}+\frac{\sigma^2_u+\delta^2}{L_u^2})$.

---

> > ### Author Response · Authors · 2023-11-22
> > **Reply to Reviewer A4M3(3).**
> >
> > ## Q1: What is the difference/advantage compared with Theorem 1 as you introduce SAM into shared parameters?
> > >Could you give more explanation on Theorem 2? What is the difference/advantage compared with Theorem 1 as you introduce SAM into shared parameters?
> > - The complete final bound in Theorem 1 is $\frac{1}{T}\sum_{i=1}^T \bigl(\frac{1}{L_u} E \bigl[\Delta_{\bar{u}}^t \bigr] + \frac{1}{L_v} E [\Delta_{v}^t \bigr] \bigr)  \leq \mathcal{O}\Big(\frac{F(\bar{u}^1, V^1) - F^*}{\sqrt{T}} + \frac{\sigma_v^2(L_v+1)}{L_v^2\sqrt{T}} + \frac{(L_{vu}^2+1)(\sigma_u^2+\delta^2)}{L_u^2 (1-\lambda)^2\sqrt{T}}+\frac{\sigma_u^2+\delta^2}{K_u L_u \sqrt{T}}+  \frac{\sigma_u^2+\delta^2}{K_u L_u(1-\lambda)^2 T} \Big)$ in Formula (20). And complete final bound in Theorem 2 is $\frac{1}{T}\sum_{i=1}^T \bigl(\frac{1}{L_u} E \bigl[\Delta_{\bar{u}}^t \bigr] + \frac{1}{L_v} E [\Delta_{v}^t \bigr] \bigr)
> >         \leq \mathcal{O}\Big(\frac{F(\bar{u}^1, V^1) - F^*}{\sqrt{T}}  + \frac{\sigma_v^2(L_v+1)}{L_v^2\sqrt{T}} +   \frac{L^2_{vu}}{(1-\lambda)^2 \sqrt{T}}\Big( \frac{\rho^2}{K_u} + \frac{\sigma_u^2+\delta^2}{L_u^2} \Big) + \frac{L_u^2 \rho^2 + \sigma_u^2+\delta^2}{L_u K_u \sqrt{T}} + \frac{L_u}{(1\!-\!\lambda)^2 T}\Big(\frac{\rho^2}{K_u} + \frac{\sigma_u^2+\delta^2}{L_u^2}\Big) \Big)$ in Formular (34). SAM introduce the gradient perturbation $\rho$ in Theorem 2.
> > - From the comparison, the dominant terms are the same in Theorem 1 and Theorem 2, which means SAM doesn't improve the convergence bound in theoretical analysis. However, it makes the shared models more robust and flatter, which will benefit the average process for each client.
> > ## Q2: Provide baseline results with more hyperparameter settings.
> > Please check the reply in M1.
> > ## Q3: Provide more details about the experiment settings.
> > Please check more details about the experiment settings in 5.1 (EXPERIMENT SETUP) and Appendix B.2 \& B.3 in the revision paper.

---

> ### Author Response · Authors · 2023-11-23
> **Last Time Reminder & Sincerely Reply to Reviewer A4M3.**
>
> Dear Reviewer A4M3,
>
> We sincerely appreciate the time and effort you have invested in reviewing our work. As the deadline for the discussion period is approaching, we kindly request that you inform us of any remaining questions you may have.
>
> We are confident that our response has adequately addressed your concerns, and we would be grateful for your feedback. Should you require further clarification, we would be delighted to answer any additional questions and provide more information.
>
> Best wishes,
>
> The Authors

---

### Author Response · Authors · 2023-11-22
**Background, Motivations and Contributions.**

## Background about DFL:
- **DFL is different from both CFL and decentralized/distributed training.**
    - **VS. CFL.** DFL discards the central server to avoid the communication congestion problem and privacy leakage risk of the central server. DFL also has some unique problems: (i) severe model inconsistency problem is caused by locality aggregation without a central server, which may cause severe over-fitting; (ii)the network connection topology has a great impact on model training, especially on heterogeneous data or sparse communication; (iii)the analysis of the specific impact of the topology (measured by the spectral gap 1-$\lambda$ of gossip matrix $W$) is more complex than CFL.
    - **VS. decentralized/distributed training.** One-step local iteration is used in decentralized/distributed training. But in general FL, clients adopt multi-step local iterations to relieve the communication burden.  The technical difficulty is that local updates fail to be an unbiased gradient estimation after multiple local iterates, thereby merging the multiple local iterations is non-trivial.
- **The advantage of DFL:**
     - Enhances decentralization by facilitating model aggregation at multiple nodes, drastically reducing dependence on a single central server and enabling direct, pairwise model sharing;
     - Employs asynchronous communications, which contributes to system resilience and ensures continued learning even if encountering nodes' delays or disconnections;
     - Local information transmissions allow for feeding the models of adjacent clients, generating a more intelligent private collaborative model.
- **Challenges for Decentralized PFL(DPFL):**
    - Effect of network topology on decentralized SGD;
    - The theoretical analysis of schemes that perform several local updates than those using a single SGD step;
    - Overfitting and insufficient generalization of the model.
## Motivations:
- Based on the advantages above, the study of PFL on DFL is meaningful.
- We reckon training and aggregation with a full model in most of the existing DPFL works lead to over-fitting in local rounds but “catastrophic forgetting" in global aggregation, which is one of the most important challenges in DPFL under data heterogeneity. Based on the works in model decoupling and partial model personalization, we consider that partial model personalized can alleviate the over-fitting problem due to the inexistence of the central server and make it outperform the centralized PFL methods under the same limited communication bandwidth.
## Contributions:
Although both model decoupling for centralized PFL and SAM for DFL have been studied independently, the contributions in our work are non-trivial and beneficial to the PFL community. In this paper, we solved the following problems and proposed an insight into improving the partial model personalization :
- We first analyze that the full model training and aggregation may lead to over-fitting in local rounds but “catastrophic forgetting" in global aggregation in DPFL. Then we propose the first extensible decentralized partial personalization framework DFedMDC, which keeps the linear classifiers locally and exchanges the shared representation in a peer-to-peer manner. It can alleviate the over-fitting problem due to the inexistence of the central server, and achieve better convergence and performance than the centralized PFL methods under the same limited communication bandwidth.
- We further explore how to improve the partial personalization in DPFL and proposed DFedSMDC, which adds proper perturbation in the gradient direction to alleviate the shared model inconsistency across clients. It demonstrates that enhancing the consistency and generalization of the shared representation parameters and keeping the classifier fitting the local distribution can both improve the personalized performance in DPFL. The insight inspires us to improve the model decoupling methods for PFL no matter in CFL or DFL via improving the shared model consistency, enhancing the personalized classifiers' adaptability to local distribution, and increasing the alignment between them. The insight and the try are non-trivial.
- We give the first convergence analysis of the partial model personalization in DPFL and carefully consider the ill impact of the statistical heterogeneity $\delta$, the smoothness $L_u, L_v, L_{vu}, L_{uv}$ of loss functions and the communication topology $(1 − \lambda)$. This is the first detailed exploration of partial personalization in DPFL, which fully considers the special convergence problems in DFL. The analysis contribution is non-trivial.
- We conduct extensive experiments to prove the effectiveness of DFedMDC and DFedSMDC under different data heterogeneity and different communication topologies, which is tightly associated with our convergence analysis.

---

### Meta-Review · Area_Chair_aV7n · 2023-12-23

**Metareview:**

Three out of four reviewers recommended rejection, one recommended acceptance. The reviewers did not change their mind after rebuttal and discussion. I have read all reviews and discussion. Moreover, I have read the paper myself. Here are some remarks:

1) The authors claim that "Theoretically, we give the first convergence analysis of the partial model personalization in DPFL, which achieves the best convergence speed in DPFL among the SOTA decentralized methods. Moreover, we first reveal the relationship between the personalized convergence and decentralized communication topology, which carefully considers the special convergence problems in DFL and it will be the basis for convergence analysis in DPFL. The convergence analysis contribution is non-trivial." However, decentralized PFL reduces to centralized (= distributed but not decentralized) PFL for a fully connected network, and I do not see any comparison of the new results obtained in this paper and prior theoretical results in the distributed setting. I believe this is absolutely necessary since without such a comparison, I find it impossible/hard to appreciate these more general results. Indeed, generalized results need to be compared and appreciated in the context of the results they are generalizing. The authors already cited some works on partial model personalization (e.g., FedSim and FedAlt). These works seem to be missing though:

- Hanzely et al, Federated learning of a mixture of global and local models, arXiv:2002.05516
- Mishchenko et al, Partially Personalized Federated Learning: Breaking the Curse of Data Heterogeneity, https://arxiv.org/abs/2305.18285
- Hanzely et al, Lower bounds and optimal algorithms for personalized federated learning, NeurIPS 2020
- Hanzely et al, Personalized federated learning: A unified framework and universal optimization techniques, https://arxiv.org/abs/2102.09743

How do your results compare to these works? In particular, the last work provides minimax optimal rates for CPFL. In particular, do your results recover these results as a special case? I believe this is not the case, and this points to a problem with the current work. In other words, I find it hard to appreciate a generalization which is not tight (unless the authors give very clear and convincing reasons for why tightness can't be obtained).

2) The obtained results give $O(1/\sqrt{K})$ rate even in the noise-less regime, i.e., when $\sigma_u^2 = \sigma_v^2 = 0$. However, in this regime, it is possible to obtain $O(1/K)$ rate for decentralized non-personalized gradient-based methods. For example, see Zhao et al, BEER: Fast O(1/T ) Rate for Decentralized Nonconvex Optimization with Communication Compression, NeurIPS 2022. Since local steps (use in the reviewed paper) are an alternative communication saving strategy to communication compression (used in BEER), it makes sense to check whether the proposed method can beat BEER in communication efficiency. It seems it can't. Moreover, the complexity should improve with personalized FL. This is not the case here. Again, I find it hard to appreciate these results.

3) Assumption 2 is too strong. For example, it is not necessarily satisfied when subsampling, which is the key algorithmic way of obtaining stochastic gradients. Due to these reasons, more modern works do not rely on this assumption. Example: Hanzely et al, Personalized federated learning: A unified framework and universal optimization techniques, https://arxiv.org/abs/2102.09743 (see their Assumption 5).  A reference focusing on explaining this phenomenon: Khaled and Richtarik, Better theory for SGD in the nonconvex world, TMLR 2022. It seems the only mechanism guaranteeing Assumption 2 holds for stochastic gradients is this: compute the gradient first, and then add zero mean bounded variance noise. Obviously, this kind of a model is too simplistic and does not capture the way stochasticity appears in applications.

4) I also believe Assumption 3 is too strong, despite the fact that the authors say "The above assumptions are mild and commonly used in the convergence analysis of FL". Indeed, many works use such assumptions. However, this does not make them weak. The reason why this assumption is strong is this: it is possible to obtain better rates than the ones you obtain in your work even without this assumption (just like gradient descent does not need it). The state-of-the-art theoretical results in CFL and CPFL do not use this assumption.

---

I could add a few other observations made by the reviewers to this list.

In summary, I do not believe the results are interesting / strong enough to be published.

**Justification For Why Not Higher Score:**

Due to the issues I mentioned above, the paper can't be accepted, so I can't suggest a higher score than a "reject".

**Justification For Why Not Lower Score:**

N/A

---

### Decision · Program_Chairs · 2024-01-16

Reject